# Bayesian Offline-to-Online Reinforcement Learning : A Realist Approach

## Abstract

Offline reinforcement learning (RL) is crucial for real-world applications where exploration can be costly. However, offline learned policies are often suboptimal and require online finetuning. In this paper, we tackle the fundamental dilemma of offline-to-online finetuning: if the agent remains pessimistic, it may fail to learn a better policy, while if it becomes optimistic directly, performance may suffer from a sudden drop. We show theoretically that the agent should adopt neither optimistic nor pessimistic policies during the offline-to-online transition. Instead, we propose a Bayesian approach, where the agent acts by sampling from its posterior and updates its belief accordingly. We demonstrate that such an agent can avoid a sudden performance drop while still being guaranteed to find the optimal policy. Based on our theoretical findings, we introduce a novel algorithm that outperforms existing methods on various benchmarks, demonstrating the efficacy of our approach. Overall, the proposed approach provides a new perspective on offline-to-online finetuning that has the potential to enable more effective learning from offline data.

## 1 Introduction

Reinforcement learning (RL) has shown impressive success in solving complex decision-making problems such as board games (Silver et al., 2016) and video games (Mnih et al., 2013), and has been applied to many real-world problems like plasma control (Degrave et al., 2022), and human preference alignment (Ouyang et al., 2022). However, RL algorithms often rely on a significant amount of exploration, which can be time-consuming and expensive. Offline RL (Levine et al., 2020) tackles such a problem by utilizing previously collected data and has gained increasing attention in recent years, with the potential to leverage large-scale and diverse datasets (Kumar et al., 2022). However, offline learned policies can be suboptimal and generalize poorly due to insufficient data and hallucination, necessitating further online fine-tuning.

To address this challenge, a hybrid approach (Nair et al., 2020; Lee et al., 2022; Song et al., 2022) has been proposed, enabling sample-efficient learning

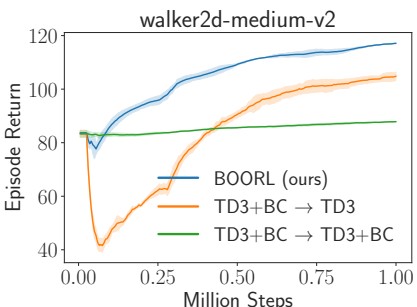

Figure 1: Fine-tuning dilemma in offline-to-online setting. Pessimistic offline methods have a slow performance improvement (green), while optimistic agents suffer from initial performance drop (orange). We develop a Bayesian-based approach to attain a fast improvement with a smaller performance drop (blue).

utilizing both previously collected data and online environments. However, previous methods do not fully address the fundamental dilemma in offline-to-online (off-to-on) RL. That is, if the algorithm remains pessimistic as it does in offline algorithms, the agent learns slowly due to a lack of exploration. Conversely, when the algorithm is optimistic, the agent's performance may suffer from a sudden drop due to inefficient use of offline knowledge and radical exploration, as shown in Figure 1. This naturally leads to the question:

*Can we design a offline-to-online algorithm that can effectively leverage offline data while exploring efficiently in a principled way?*

To answer this question, we integrate information-theoretic concepts into the design and analysis of RL algorithms. Our results show that the Bayesian approach has strong guarantees and is superior over both optimism (e.g., UCB) and pessimism (e.g., LCB) in off-to-on settings. Intuitively, by sampling from the posterior rather than taking the most optimistic one or the most pessimistic one, it achieves a balance between reusing known experiences and exploring the unknowns. We derive a concrete bound in linear MDPs and conduct experiments in didactic bandits to further demonstrate the superiority of the Bayesian approach in off-to-on settings. Based on the theoretical results, we design an efficient offline-to-online algorithm by leveraging the idea of bootstrapping (Osband et al., 2016). Experiments show that our algorithm effectively resolves the dilemma, which effectively explores while avoiding a sudden drop in performance. Also, our algorithm is generally compatible with off-the-shelf offline RL methods for off-to-on transition.

Our contribution is threefold: (1) we provide an information-theoretic characterization of RL algorithms' performance that links online and offline performance with the agent's gained information about the environment, (2) we demonstrate the superiority of the Bayesian approach in offline-to-online RL theoretically, and (3) we develop a practical approach with bootstrapping for offline-to-online RL and achieve superior performance on various tasks. Overall, our proposed approach provides a new perspective on offline-to-online fine-tuning that has the potential to enable more effective learning from offline data.

### 1.1 RELATED WORKS

**Offline-to-Online RL.**   On the empirical side, Nair et al. (2020) is among the first to propose a direct solution to off-to-on RL. Prior works like(Lee et al., 2022; Zhang et al., 2023) propose various approaches, including balanced relay buffer and policy expansion, to reuse offline knowledge more efficiently. Nakamoto et al. (2023) observes an optimistic-pessimistic dilemma similar to ours and proposes calibrating the offline and online learned value function. However, they do not formally point out such a dilemma nor analyze it in a principled way.

On the theoretical side, Xie et al. (2021) shows the importance of online exploration when the offline dataset only has partial coverage. Song et al. (2022) demonstrates cases where a purely offline dataset can fail while a hybrid approach succeeds, and Xie et al. (2022) shows an interesting connection between offline concentration coefficient and online learning efficiency.

**Bayesian RL and Information-Theoretic Analysis.**   Osband & Van Roy (2017); Russo & Van Roy (2014) theoretically justify the effectiveness of Bayesian methods like Thompson sampling. Uehara & Sun (2021) analyzes the performance of Bayesian methods in the offline setting. Lu & Van Roy (2019) derives an information-theoretical formulation to analyze the regret bound of online learning algorithms like UCB and TS. Our work extends their work to offline and off-to-on settings.

On the empirical side, (Osband et al., 2016) first adopts a Bayesian view into the exploration of deep RL. (Chua et al., 2018) proposes a model-based approach for Bayesian exploration. Ghosh et al. (2022) adopts the Bayesian principle in the offline setting. Our work extends these works to the off-to-on setting.

## 2 PRELIMINARIES

### 2.1 EPISODIC REINFORCEMENT LEARNING

We consider finite-horizon episodic Markov Decision Processes (MDPs), defined by the tuple $(\mathcal{S}, \mathcal{A}, H, \mathcal{P}, r)$, where $\mathcal{S}$ is a state space, $\mathcal{A}$ is an action space, $H$ is the horizon and $\mathcal{P} = \{\mathcal{P}_h\}_{h=1}^H, r = \{r_h\}_{h=1}^H$ are the transition function and reward function, respectively.

A policy $\pi = \{\pi_h\}_{h=1}^H$ specifies a decision-making strategy in which the agent chooses its actions based on the current state, i.e., $a_h \sim \pi_h(\cdot \,|\, s_h)$. The value function $V_h^\pi : \mathcal{S} \to \mathbb{R}$ is defined as the

sum of future rewards starting at state $s$ and step $h \in [H]$, and similarly, the Q-value function, i.e.

$$V_h^\pi(s) = \mathbb{E}_\pi\Big[\sum_{t=h}^{H} r_t(s_t, a_t) \Big| s_h = s\Big], \quad Q_h^\pi(s, a) = \mathbb{E}_\pi\Big[\sum_{t=h}^{H} r_h(s_t, a_t) \Big| s_h = s, a_h = a\Big]. \quad (1)$$

where the expectation is w.r.t. the trajectory $\tau$ induced by $\pi$. We define the Bellman operator as

$$(\mathbb{B}_h f)(s, a) = \mathbb{E}\big[r_h(s, a) + f(s')\big], \quad (2)$$

for any $f : \mathcal{S} \to \mathbb{R}$ and $h \in [H]$. The optimal Q-function $Q^*$, optimal value function $V^*$ and optimal policy $\pi^*$ are related by the Bellman optimality equation

$$V_h^*(s) = \max_{a \in \mathcal{A}} Q_h^*(s, a), \quad Q_h^*(s, a) = (\mathbb{B}_h V_{h+1}^*)(s, a), \quad \pi_h^*(\cdot \mid s) = \operatorname*{argmax}_\pi \mathbb{E}_{a \sim \pi} Q_h^*(s, a). \quad (3)$$

We define the suboptimality, or the per-episode regret as the performance difference of the optimal policy $\pi^*$ and the current policy $\pi_k$ given the initial state $s_1 = s$. That is

$$\Delta_k = \mathrm{SubOpt}(\pi_k; s) = V_1^{\pi^*}(s) - V_1^{\pi_k}(s).$$

## 2.2 Linear Function Approximation

To derive a concrete bound for Bayesian offline-to-online learning, we consider the *linear MDP* (Jin et al., 2020; 2021) as follows, where the transition kernel and expected reward function are linear with respect to a feature map, which indicate that the value function is also linear.

**Definition 2.1** (Linear MDP). $\mathrm{MDP}(\mathcal{S}, \mathcal{A}, \mathrm{H}, \mathbb{P}, \mathrm{r})$ is a *linear MDP* with a feature map $\phi : \mathcal{S} \times \mathcal{A} \to \mathbb{R}^d$, if for any $h \in [H]$, there exist $d$ *unknown* (signed) measures $\mu_h = (\mu_h^{(1)}, \ldots, \mu_h^{(d)})$ over $\mathcal{S}$ and an *unknown* vector $\theta_h \in \mathbb{R}^d$, such that for any $(s, a) \in \mathcal{S} \times \mathcal{A}$, we have

$$\mathbb{P}_h(\cdot \mid s, a) = \langle \phi(s, a), \mu_h(\cdot) \rangle, \qquad r_h(s, a) = \langle \phi(s, a), \theta_h \rangle. \quad (4)$$

Without loss of generality, we assume $||\phi(s, a)|| \leq 1$ for all $(s, a) \in \mathcal{S} \times \mathcal{A}$, and $\max\{||\mu_h(\mathcal{S})||, ||\theta_h||\} \leq \sqrt{d}$ for all $h \in [H]$.

## 2.3 Information Gain and Bayesian Learning

Let $\mathcal{H}_{k,h} = (s_{1,1}, a_{1,1}, r_{1,1}, \ldots, s_{k,h-1}, a_{k,h-1}, r_{k,h-1}, s_{k,h})$ be all the history up to step $h$ of episode $k$. We use subscript $k, h$ to indicate quantities conditioned on $\mathcal{H}_{k,h}$, i.e. $\mathbb{P}_{k,h} = \mathbb{P}(\cdot|\mathcal{H}_{k,h}), \mathbb{E}_{k,h}[\cdot] = \mathbb{E}[\cdot|\mathcal{H}_{k,h}]$. The filtered mutual information is defined as

$$I_{k,h}(X; Y) = D_{\mathrm{KL}}(P_{k,h}(X, Y) || P_{k,h}(X) P_{k,h}(Y)),$$

which is a random variable of $\mathcal{H}_{k,h}$. For a horizon dependent quantity $f_{k,h}$, we define $\mathbb{E}_k[f_k] = \sum_{h=1}^{H} \mathbb{E}_{k,h}[f_{k,h}]$ and similarly for $\mathbb{P}_k$. We use $t$ instead of $k, h$ for simplicity when it does not lead to confusion, e.g., $I_t \triangleq I_{k,h}$.

We also define the information ratio (Russo & Van Roy, 2016) as the ratio between the expected single step regret and the expected reduction in entropy of the unknown parameter as follows

**Definition 2.2** (Information Ratio). The information ratio $\Gamma_t$ given history $\mathcal{H}_t$ is the supremum value $\Gamma$ such that

$$\mathbb{P}_k\left(|Q_w(s, a) - \mathbb{E}_k Q_w(s, a)| \leq \frac{\Gamma}{2}\sqrt{I_t(w_h; r_{t,a}, s_{t+1,a})}, \forall h \in [H], s \in \mathcal{S}, a \in \mathcal{A}\right) > 1 - \frac{\delta}{2}.$$

From a Bayesian point of view, we assume that the MDP can be described by an unknown model parameter $w = \{w_h\}_{h=1}^{H}$, which governs the outcome distribution. The agent's belief over the environment at the $t$-th timestep is represented as a distribution $\beta_t$ over $w$. We reload $\pi$ as an algorithm that generates a sequence of functions $\{\pi\}_{k=1}^{K}$ that map histories and current states to distributions over actions. We define the Bayesian regret of an algorithm $\pi$ over $T$ periods

$$\mathrm{BayesRegret}(T, \pi) = \mathbb{E}[\mathrm{Regret}(T, \pi)] = \sum_{k=1}^{K} \mathbb{E}_{k, \beta_k}[\Delta_k],$$

where $T = HK$ and the expectation is taken over the randomness in outcomes, algorithm $\pi$, as well as the posterior distribution $\beta_{k,h}$ over $w$. We also use BayesRegret($N, T, \pi$) to denote the offline-to-online regret of an algorithm $\pi$ that uses an offline dataset of size $N = LH$ and interacts online for $T = HK$ steps.

Similar to the definition of the concentration coefficient in offline RL literature (Jin et al., 2021; Uehara & Sun, 2021), we can generalize such a concept by taking the expectation over the belief $\beta$. Specifically, we have the following definition (Uehara & Sun, 2021).

**Definition 2.3.** The Bayesian concentration coefficient with respect to the feature map $\phi(s, a)$ and posterior $\beta$ is defined as

$$C_\beta^\dagger = \max_{h \in [H]} \mathbb{E}_{w \sim \beta} \sup_{\|x\|=1} \frac{x^\top \Sigma_{\pi_w^*, h} x}{x^\top \Sigma_{\rho_h} x}, \tag{5}$$

where $\Sigma_{\pi_w^*, h} = \mathbb{E}_{(s,a) \sim d_{\pi_w^*, h}(s,a)}[\phi(s, a)\phi(s, a)^\top], \Sigma_{\rho_h} = \mathbb{E}_{\rho_h}[\phi(s, a)\phi(s, a)^\top]$.

Bayesian concentration coefficient is a natural generalization of normal concentration coefficient (Uehara & Sun, 2021; Jin et al., 2021; Rashidinejad et al., 2021) in Baysian settings and has appeared in previous work (Uehara & Sun, 2021).

## 3 THEORETICAL ANALYSIS

It is known that we should adopt optimistic algorithms (e.g., UCB (Auer, 2002)) in online settings to avoid missing optimal strategies, and we should adopt pessimistic algorithms (e.g., LCB (Rashidinejad et al., 2021)) to avoid overconfidence in unknown regions. However, it is unclear what is the principled way for offline-to-online settings where both an offline dataset and an online environment are available. As Figure 1 demonstrates, optimistic online algorithms (e.g., TD3 (Fujimoto et al., 2018)) can mismanage prior knowledge in the dataset, leading to a sudden drop in performance. On the other hand, pessimistic offline algorithms (e.g., TD3+BC (Fujimoto & Gu, 2021)) can be too conservative in exploration, which leads to slow learning.

We conduct an information-theoretic analysis with a Bayesian point of view in Section 3.1 to understand how we can use both the dataset and the environment properly. Specifically, we cast the dataset as priors and the online interaction as updating posteriors. From such a point of view, we show that optimistic algorithms like UCB can utilize their posterior to make quick adaptations, and pessimistic algorithms like LCB can utilize their posterior to avoid risky trials, which aligns with prior findings. More interestingly, we show that a Bayesian agent (e.g., Thompson Sampling; TS) can utilize its posterior to do both and outperform optimistic and pessimistic agents. Intuitively, uniformly sampling from the posterior rather than acting according to the most optimistic or the most pessimistic estimation strikes a proper balance between efficient exploration and safe exploitation. Such property leads to a concrete performance bound for Bayesian agents in offline-to-online settings with linear MDPs, which is probably better than UCB and LCB agents as illustrated in Section 3.2. Such theoretical prediction matches well with empirical observations on didactic bandit settings, as shown in Figure 2. Overall, our insight is concluded in Table 1, indicating that we should adopt neither optimism nor pessimism in the offline-to-online setting but a "realist" approach that samples from the posterior uniformly.

### 3.1 INFORMATION-THEORETIC ANALYSIS

What are good exploration and exploitation strategies, information-theoretically? Lu & Van Roy (2019) gives a nice answer for the case of online exploration. That is, a good exploration strategy incurs a suboptimality only when it can learn a lot from the environment. Therefore, the suboptimality at each step should be proportional to the possible information gain. Similarly, for offline exploitation, a good exploitation strategy should incur a suboptimality only due to its uncertainty about the environment after learning from the offline dataset. This allows us to redefine abstract exploration and exploitation strategies like UCB, LCB, and TS in an abstract and information-theoretic manner, with details shown in Appendix A.1. For the above abstract algorithms, we have the following performance guarantees.

**Theorem 3.1.** *Suppose*

$$\mathbb{P}_k \left( \left| Q_{t,w}(s,a) - \bar{Q}_{t,w}(s,a) \right| \leq \frac{\Gamma_t}{2} \sqrt{I_t(w_h; r_{t,a}, s_{t+1,a})}, \forall h \in [H], s \in \mathcal{S}, a \in \mathcal{A} \right) \quad (6)$$

*is greater than $1 - \delta/2$, where $\bar{Q}_{t,w}(s,a)$ is the Bayesian average value, i.e., $\bar{Q}_{t,w}(s,a) = \mathbb{E}_{w \sim \beta_t}[Q_{t,w}(s,a)]$.*

*Then the per-episode regret of Thompson Sampling and UCB agents satisfies*

$$\mathbb{E}_k[\Delta_k] \leq \sum_{h=1}^{H} \Gamma_t \mathbb{E}_k \left[ \sqrt{I_t(w_h; a_t, r_t, s_{t+1})} \right] + 2\delta H^2. \quad (7)$$

*Similarly, the per-episode regret of Thompson Sampling and LCB agents satisfies*

$$\mathbb{E}_k[\Delta_k] \leq \sum_{h=1}^{H} \Gamma_t \mathbb{E}_{\pi^*} \left[ \sqrt{I_t(w_h; a_t^*, r_t, s_{t+1})} \right] + 2\delta H^2. \quad (8)$$

*Proof.* Please refer to Appendix C.1 for detailed proof. □

Equation (6) abstracts away specific structures of MDPs (e.g., linear MDP, etc.) and only assumes that the uncertainty in the Q-value function can be reduced at a certain rate as we gain more information about the environment. Equation (6) generally holds in various settings, including linear MDPs (Jin et al., 2020), factored MDPs (Lu & Van Roy, 2019), and kernel MDPs (Yang & Wang, 2020). Please refer to Lu & Van Roy (2019) for a detailed derivation.

Theorem 3.1 leads to an information-theoretic performance bound. Equation (7) indicates an online $\widetilde{\mathcal{O}}(\sqrt{T})$-regret bound using the chain rule of mutual information, as depicted in Proposition B.1. With additional assumption on the coverage of the dataset, Equation (8) implies an $\widetilde{\mathcal{O}}(\sqrt{C/N})$ offline performance bound where $C$ is the coverage coefficient. Note that Thompson sampling enjoys both regret bounds in Equation (7) and Equation (8), which indicates that Thompson sampling is suitable for both offline and online settings. Moreover, it indicates that a Bayesian approach enjoys better guarantees in offline-to-online settings since it can avoid sudden performance drop (due to Equation (8)) and explore efficiently (due to Equation (7)). This is summarized in Table 1, where we provide a classification of existing settings and corresponding doctrines. Table 1 suggests that the Bayesian approach is consistent across different settings and recommends a realist approach in offline-to-online settings, as opposed to optimism or pessimism.

| Setting | Doctrine | Algorithm |
|---|---|---|
| Online Learning | Optimism | TS, UCB |
| Offline Learning | Pessimism | TS, LCB |
| Offline-to-online | Realism | TS |

Table 1: A taxonomy of the doctrines in different settings of reinforcement learning. a Bayesian approach like TS is generally suitable for online, offline and offline-to-online settings, and is the only one that works in the offline-to-online setting.

## 3.2 SPECIFICATION IN LINEAR MDPS

In this section, we provide specific regret bounds for Bayesian methods in linear MDPs when both offline data and online interactions are available. Applying Theorem 3.1 to linear MDPs as defined in Definition 2.1, we have the following theorem.

**Theorem 3.2** (Regret of Bayesian Agents in Linear MDPs, informal). *Given an offline dataset $\mathcal{D}$ of size $N$, the regret of Thompson sampling during online interaction satisfies the following bound:*

$$\text{BayesRegret}(N, T, \pi) \leq c\sqrt{d^3 H^3 \iota} \left( \sqrt{\frac{N}{C_\beta^\dagger} + T} - \sqrt{\frac{N}{C_\beta^\dagger}} \right), \quad (9)$$

*where $\iota$ is a logarithmic factor and $c$ is an absolute constant.*

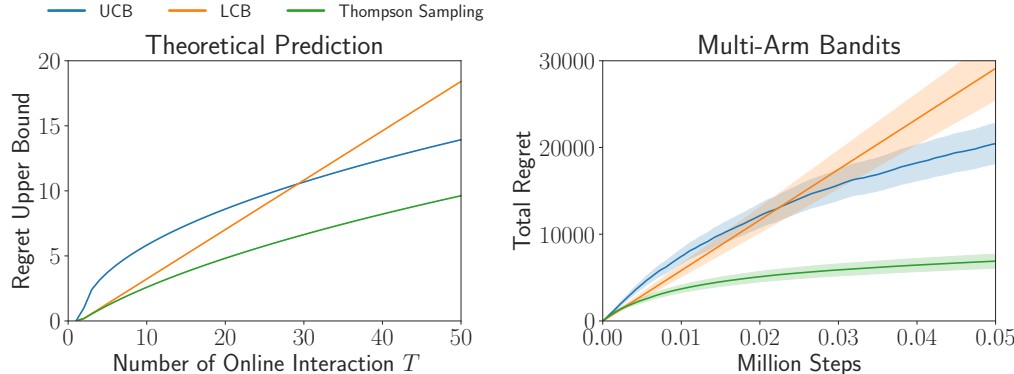

Figure 2: Theoretical prediction of Theorem 3.2 and experiments on multi-arm bandits. The performance of a Bayesian approach matches the performance of LCB at an early stage by using prior knowledge in the dataset properly and matches the performance of UCB in the run by allowing efficient exploration. Therefore, a Bayes agent performs better than both UCB and LCB agents. Experiments on multi-arm bandits match well with our theoretical prediction.

*Proof.* Please refer to Appendix C.2 for detailed proof. ☐

Theorem 3.2 demonstrates that the Bayesian approach provides a robust regret guarantee. From simple algebraic observations that $\sqrt{a+b} - \sqrt{a} \leq \sqrt{b}$ and $\sqrt{a+b} - \sqrt{a} \leq b/(2\sqrt{a})$, Theorem 3.2 indicates that Bayes agent can have a jump start with low regret (i.e., $\widetilde{\mathcal{O}}(\sqrt{C_\beta^\dagger/N})$) and converges to the optimal policy at an $\widetilde{\mathcal{O}}(\sqrt{T})$ rate, a feat neither naive online nor offline approaches can accomplish alone. This is further formalized in Propositions 3.3 and 3.4. To further verify our theoretical findings, we conducted an experiment on didactic bandit settings; the result is shown in Figure 2. Experiment results align well with our theoretical predictions in Equation 15. Please refer to Appendix E for more experiment details on the didactic bandits.

**Proposition 3.3.** *Under the same assumption of Theorem 3.2, the expected one-step suboptimality of UCB can be unbounded (i.e. $\widetilde{O}(1)$), while the expected suboptimality of Thompson sampling satisfies*

$$\text{SubOpt}(N, T, \pi) \leq c\sqrt{\frac{C_\beta^\dagger d^3 H^3 \iota}{N}} = \widetilde{\mathcal{O}}\left(\sqrt{\frac{C_\beta^\dagger d^3 H^3}{N}}\right).$$

**Proposition 3.4.** *Under the same assumption of Theorem 3.2, the regret of LCB can be unbounded (i.e. $\widetilde{O}(T)$), while the regret of Thompson sampling satisfies*

$$\text{BayesRegret}(N, T, \pi) \leq 2c\sqrt{d^3 H^3 T \iota} = \widetilde{\mathcal{O}}(\sqrt{d^3 H^3 T}).$$

Theorem 3.2 is a significant departure from previous findings. Xie et al. (2021) analyzes the benefit of online exploration when offline data only has partial coverage, while Song et al. (2022) proposes Hy-Q which has an oracle property ($O(\sqrt{T})$ regret compared with **any** policy $\pi_e$) and is computationally efficient by incorporating offline data. Different from previous result, our result shows that using offline data with full coverage can improve over the $O(\sqrt{T})$ online regret bound by adopting a Bayesian method. Moreover, our performance bound incorporates both the number of online interactions $T$ and the offline dataset size $N$, demonstrating that both elements play a key role in minimizing (amortized) regret.

## 4 ALGORITHM

Based on the theoretical analysis in Section 3, we propose a simple yet effective Bayesian Offline-to-Online Reinforcement Learning (BOORL) method to address the dilemma in offline-to-online RL. The algorithm procedure is shown in Appendix A.2.

## 4.1 OFFLINE-TO-ONLINE THOMPSON SAMPLING

We adopt the bootstrapped mechanism as the natural adaptation of the Thompson Sampling heuristic to offline-to-online RL. In the offline phase, we modify the offline RL algorithm to approximate a distribution over policy via the bootstrap. Specifically, we randomly initialize $N$ policy networks and corresponding $Q$-value networks $\{\pi_{\phi_i}, Q_{\theta_i}\}_{i=1}^N$. We use the independent mask $m_1, m_2, \cdots, m_N \in M$ for each policy to implement an offline bootstrap. These flags are stored in the memory replay buffer $\mathcal{D}_i^{\text{off}}$ and identify which policies are trained on which data. Next, each one of these offline policies is trained against its own pessimistic $Q$-value network and bootstrapped dataset with the offline RL loss (e.g., TD3+BC (Fujimoto & Gu, 2021)):

$$\mathcal{L}_{\text{critic}}(\theta_i) = \mathbb{E}_{(s,a,r,s',m_i) \sim \mathcal{D}_i^{\text{off}}} \left[ \left( r + \gamma Q_{\theta_i'}(s', \widetilde{a}) - Q_{\theta_i}(s,a) \right)^2 * m_i \right] \tag{10}$$

$$\mathcal{L}_{\text{actor}}(\phi_i) = -\mathbb{E}_{(s,a,m_i) \sim \mathcal{D}_i^{\text{off}}} \left[ \left( \lambda Q_{\theta_i}(s, \pi_{\phi_i}(s)) - (\pi_{\phi_i}(s) - a)^2 \right) * m_i \right], \tag{11}$$

where $\widetilde{a} = \pi_{\phi_i'}(s') + \epsilon$ is the target plicy smoothing regularization and $\lambda$ is the hyper-parameter for behavior cloning. As for the online phase, we first load the offline pre-trained model. Then, we approximate a bootstrap sample by selecting $n \in \{1, \cdots, N\}$ uniformly at random at each time step and following $\pi_n$ to collect online data. Each loaded policy and $Q$-value network is continued to be trained with the online RL loss (e.g., TD3 (Fujimoto et al., 2018)):

$$\mathcal{L}_{\text{critic}}(\theta_i) = \mathbb{E}_{(s,a,r,s') \sim \mathcal{D}^{\text{off}} \cup \mathcal{D}^{\text{on}}} \left[ \left( r + \gamma Q_{\theta_i'}(s', \widetilde{a}) - Q_{\theta_i}(s,a) \right) \right]^2 \tag{12}$$

$$\mathcal{L}_{\text{actor}}(\phi_i) = -\mathbb{E}_{(s,a) \sim \mathcal{D}^{\text{off}} \cup \mathcal{D}^{\text{on}}} \left[ Q_{\theta_i}(s, \pi_{\phi_i}(s)) \right], \tag{13}$$

The method above with deep bootstrapping (Osband et al., 2016) is a natural implementation of Thompson sampling in the offline-to-online setting. Note that BOORL can also be combined with most offline methods with minor modifications.

In the online phase, sample selection is essential for fine-tuning. A naive approach is using a single replay buffer for both offline and online samples, then sampling uniformly at random, In that case, the agent does not use enough online samples for updates, especially when the large offline dataset leads to slow fine-tuning. We adopt a simple yet efficient sample selection method Ross & Bagnell (2012); Ball et al. (2023) to incorporate prior data better. For each batch, we sample $50\%$ of the data from the online replay buffer $\mathcal{D}^{\text{on}}$, and the remaining $50\%$ from the offline replay buffer $\mathcal{D}^{\text{off}}$. Further, we increase the UTD ratio $G$ to make the Bellman backups perform as sample-efficiently as possible.

The overall algorithm is summarized in Algorithm 4 and Algorithm 5 in Appendix A.2. We highlight elements important to our approach in Purple. In practice, we use the Bernoulli mask $m_1, m_2, \cdots, m_N \in \text{Ber}(p)$ to each offline policy, where $p$ is the mask ratio.

## 5 EXPERIMENTS

We design our experiments to answer the following questions: (1) Whether BOORL can effectively solve the dilemma in offline-to-online RL? (2) How does BOORL compare with other state-of-the-art approaches for finetuning pre-trained policies? (3) Whether BOORL is general and can be effectively combined with other off-the-shelf offline RL algorithms?

To answer the questions above, we conduct experiment to test our proposed approach on the D4RL benchmark (Fu et al., 2020), which encompasses a variety of dataset qualities and domains. We adopt the normalized score metric proposed by the D4RL benchmark (Fu et al., 2020), averaging over five random seeds with standard deviation.

**Answer of Question 1:** We compare BOORL with the online version of TD3+BC (Fujimoto & Gu, 2021), named TD3+BC (online), as well as directly using TD3 for finetuning, named TD3 (finetune). For the fair and identical experimental evaluation, these three methods are pre-trained based on TD3+BC for 1 million time steps and adopt the TD3 algorithm for online learning.

The results in Figure 6 in Appendix F show TD3+BC exhibits safe but slow performance improvement, resulting in worse asymptotic performance. On the other hand, TD3 suffers from initial performance

| Task | Type | ODT | Off2On | Cal-QL | BOORL |
|------|------|-----|--------|--------|-------|
| Hopper | random | 10.1→30.8 | 6.9→18.6 | 9.3→11.9 | 8.8→**75.7** |
| | medium | 66.9→97.5 | 65.8→104.6 | 75.8→100.6 | 61.9→**109.8** |
| | medium-replay | 86.6→88.8 | 89.8→106.5 | 95.4→106.1 | 75.5→**111.1** |
| | medium-expert | 107.6→**111.1** | 83.8→**111.3** | 85.0→**111.6** | 89.0→103.4 |
| | expert | 108.1→**110.7** | 109.2→92.2 | 94.8→**110.3** | 111.5→109.2 |
| Walker2d | random | 4.6→8.8 | 2.1→9.8 | 14.8→17.3 | 4.8→**93.6** |
| | medium | 72.1→76.7 | 82.1→105.6 | 80.8→89.6 | 83.6→**107.7** |
| | medium-replay | 68.9→76.8 | 81.8→104.2 | 83.8→94.5 | 69.1→**114.4** |
| | medium-expert | 108.1→108.7 | 111.2→**119.0** | 106.8→111.0 | 110.8→116.2 |
| | expert | 108.2→107.6 | 108.4→**117.6** | 108.8→109.2 | 110.0→**118.6** |
| Halfcheetah | random | 1.1→2.2 | 28.4→94.0 | 22.0→45.1 | 10.7→**97.7** |
| | medium | 42.7→42.1 | 47.8→83.3 | 48.0→72.3 | 47.9→**98.7** |
| | medium-replay | 39.9→40.4 | 46.9→88.0 | 46.5→59.5 | 44.5→**91.5** |
| | medium-expert | 86.8→94.1 | 45.3→94.4 | 48.0→90.2 | 77.7→**97.9** |
| | expert | 87.3→94.3 | 95.9→93.7 | 64.5→92.1 | 97.5→**98.4** |
| Antmaze | umaze | 55.0→85.0 | 75.0→0.0 | 80.0→**100.0** | 80.0→**100.0** |
| | medium-replay | 0.0→0.0 | 0.0→0.0 | 60.0→90.0 | 50.0→**100.0** |
| | medium-diverse | 0.0→0.0 | 0.0→0.0 | 70.0→**85.0** | 60.0→**85.0** |
| | large-play | 0.0→0.0 | 0.0→0.0 | 25.0→55.0 | 60.0→**75.0** |
| $\delta_{\text{sum}}$ (0.2M) | | 121.6 | 258.7 | 331.9 | **650.7** |

Table 2: Normalized score before and after the online fine-tuning with five random seeds. We pre-trained each method for 1M steps and then fine-tuned 0.2M environment steps. Since offline algorithms' performance differs, we focus on performance improvement within a limited time, $\delta_{\text{sum}}$ (0.2M), which denotes the sum of performance improvement on all tasks within 0.2M steps.

degradation, especially in narrow distribution datasets (e.g., expert datasets). Differently, BOORL attains a fast performance improvement with a smaller regret. Due to the offline bootstrap, the initial performance in the online phase between BOORL and baselines exits a small difference, while it does not change the conclusion.

**Answer of Question 2:** We compare BOORL with several strong offline-to-online algorithms, inclduing ODT (Zheng et al., 2022), Off2On (Lee et al., 2022), AWAC (Nair et al., 2020), PEX (Zhang et al., 2023) and Cal-QL (Nakamoto et al., 2023). We re-run the official implementation to offline pre-train for 1 million steps. Then we report the fine-tune performance for 200k online steps. As for BOORL, we use TD3+BC and TD3 as the backbone of offline and online algorithms. The results in Table 2 show that our algorithm achieves superior fine-tuning performance and notable performance improvement $\delta_{\text{sum}}$ compared with other fine-tuning approaches. The results in Figure 3 show our method achieves better learning efficiency and stability compared with these baselines. AWAC has limited efficiency due to a lack of online adaptability, and PEX is not as stable as ours. The concurrent work, Cal-QL, achieves comparable stability due to calibration, but our method demonstrates better sample efficiency in general.

**Answer of Question 3:** We incorporate BOORL with another offline RL algorithm, IQL, and evaluate it on the sparse reward task in the D4RL benchmark, Antmaze. Consistent with the previous experimental setup, we first offline train IQL for 1 million time steps and then load the same pre-trained weight for BOORL. The experimental results in Table 2 show that BOORL achieves superior performance and higher sample efficiency than other baselines. This demonstrates that BOORL can be easily extended to various offline RL algorithms.

**Ablation Study:** To delved deeper into the performance of Bayesian methods, we enforced a strict offline → online transition. Specifically, we exclusively loaded the offline-trained policy and Q-network module, omitting the offline data during the online phase. We refer to this setup as "Thompson Sampling". Furthermore, we examined the naive offline-to-online (TD3+BC → TD3)

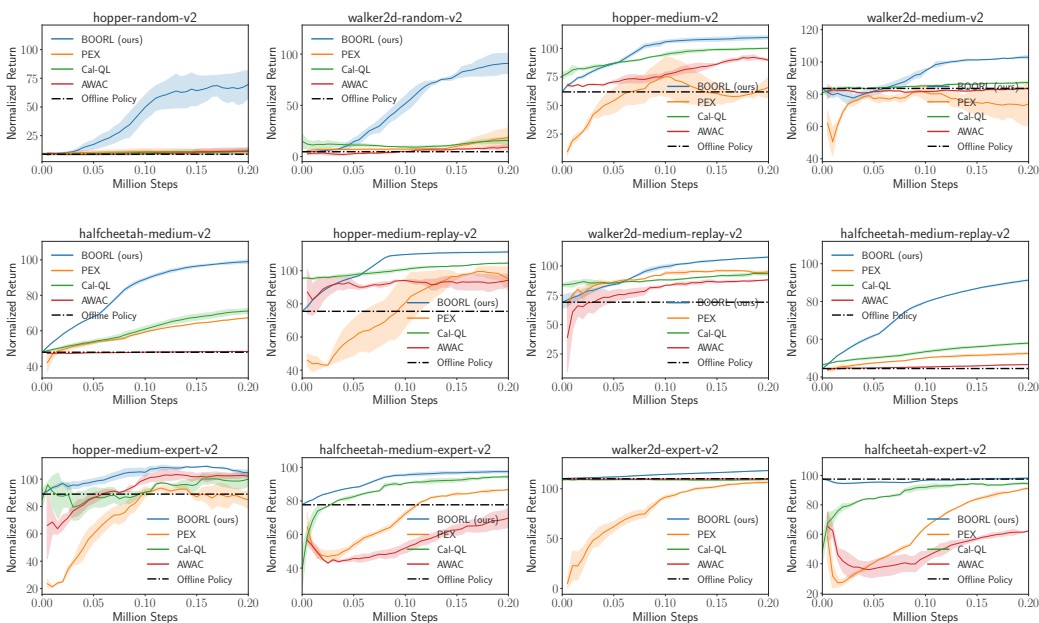

Figure 3: Experiments between several baselines and BOORL. The reference line is the performance of TD3+BC. The experimental results are averaged with five random seeds.

| Task | Type | BOORL | Thompson Sampling | $\delta$ | Hybrid RL | $\delta$ |
|---|---|---|---|---|---|---|
| | random | 75.7±1.3 | 85.4±3.3 | -9.7 | 75.2±3.9 | 0.5 |
| Hopper | medium | 109.8±1.6 | 109.6±1.5 | 0.2 | 91.4±1.2 | 18.4 |
| | medium-replay | 111.1±0.3 | 110.6±0.6 | 0.5 | 103.5±2.7 | 7.6 |
| | random | 93.6±4.4 | 92.4±4.7 | 1.2 | 15.4±0.8 | 78.2 |
| Walker2d | medium | 107.7±0.5 | 96.5±3.5 | 11.2 | 86.4±0.4 | 21.3 |
| | medium-replay | 114.4±0.9 | 103.7±2.1 | 10.7 | 99.7±2.4 | 14.7 |
| | random | 97.7±1.1 | 94.5±4.2 | 3.2 | 85.2±0.5 | 12.5 |
| Halfcheetah | medium | 98.7±0.3 | 97.7±0.5 | 1.0 | 80.3±0.2 | 18.4 |
| | medium-replay | 91.5±0.9 | 90.5±0.5 | 1.0 | 84.8±1.0 | 6.7 |

Table 3: Ablation results on Mujoco tasks with the normalized score metric, averaged over five random seeds with standard deviation.

with the Hybrid RL framework to examine the effects of integrating offline data, termed "Hybrid RL". Results in Table 3 reveal that Thompson Sampling exhibits a large performance difference in the majority of tasks. We conduct addition ablation studies to understand the behavior of BOORL. Please refer to Appendix H for detailed experimental details and results.

## 6 CONCLUSION

Our work presents a novel perspective on off-to-on RL, effectively tackling the inherent challenge of balancing exploration efficiency and utilizing offline data. Based on the information-theoretic analysis, we show that Bayesian methods can be efficaciously applied for offline-to-online transitions. By leveraging the concept of bootstrapping, our algorithm outperforms previous methods by resolving the dichotomy between exploration and performance and demonstrating superior outcomes across various tasks. It is an interesting future direction to design more efficient algorithms from the Bayesian point of view for off-to-on transition.

## 7 REPRODUCIBILITY

A comprehensive description of our algorithm implementation is provided in Section 4. The hyper-parameter configurations are detailed in Appendix J. The code necessary to reproduce BOORL is available in our supplementary materials. Our theoretical findings are expounded upon in Section 3, with a detailed proof presented in Appendix B.

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

# A  ALGORITHM DETAILS

## A.1  DETAILS OF ABSTRACT ALGORITHMS

In this section, we provide an information-theoretic abstraction of the UCB, LCB and Thompson Sampling algorithm when both offline dataset and online environment are present. Here we use $t$ to represent $(k, h)$ for simplicity.

---

**Algorithm 1** Upper Confidence Bound

---

1: **Require**: offline dataset $\mathcal{D}^{\text{off}}$, online interaction episodes $K$, exploration coefficient $\Gamma_t$,
2: Initialize prior $\beta_0$ with the offline dataset $\mathcal{D}^{\text{off}}$.
3: **for** $k = 1, \cdots, K$ **do**
4:     **for** $h = 1, \cdots, H$ **do**
5:         Calculate posterior mean $\bar{Q}_{w,t}(\cdot, \cdot) = \mathbb{E}_{w \sim \beta_t}[Q_{t,w}(\cdot, \cdot)]$.
6:         Calculate optimistic value function $\widehat{Q}(\cdot, \cdot) = \bar{Q}_{w,t}(\cdot, \cdot) + \frac{\Gamma_t}{2}\sqrt{I_t(w; \cdot, \cdot)}$.
7:         Execute $a_t = \operatorname{argmax}_{a_t} \widehat{Q}(s_t, a)$ and receive feedback $s_{t+1}, r_t$.
8:         Update posterior $\beta_t$ with evidence $(a_t, r_t, s_{t+1})$.
9:     **end for**
10: **end for**

---

---

**Algorithm 2** Lower Confidence Bound

---

1: **Require**: offline dataset $\mathcal{D}^{\text{off}}$, online interaction episodes $K$, exploration coefficient $\Gamma_t$,
2: Initialize prior $\beta_0$ with the offline dataset $\mathcal{D}^{\text{off}}$.
3: **for** $k = 1, \cdots, K$ **do**
4:     **for** $h = 1, \cdots, H$ **do**
5:         Calculate posterior mean $\bar{Q}_{w,t}(\cdot, \cdot) = \mathbb{E}_{w \sim \beta_t}[Q_{t,w}(\cdot, \cdot)]$.
6:         Calculate pessimistic value function $\widehat{Q}(\cdot, \cdot) = \bar{Q}_{w,t}(\cdot, \cdot) - \frac{\Gamma_t}{2}\sqrt{I_t(w; \cdot, \cdot)}$.
7:         Execute $a_t = \operatorname{argmax}_{a_t} \widehat{Q}(s_t, a)$ and receive feedback $s_{t+1}, r_t$.
8:         Update posterior $\beta_t$ with evidence $(a_t, r_t, s_{t+1})$.
9:     **end for**
10: **end for**

---

---

**Algorithm 3** Thompson Sampling

---

1: **Require**: offline dataset $\mathcal{D}^{\text{off}}$, online interaction episodes $K$, exploration coefficient $\Gamma_t$,
2: Initialize prior $\beta_0$ with the offline dataset $\mathcal{D}^{\text{off}}$.
3: **for** $k = 1, \cdots, K$ **do**
4:     **for** $h = 1, \cdots, H$ **do**
5:         Sample parameter $w_t$ from posterior $\beta_t$.
6:         Calculate corresponding value function $\widehat{Q}(\cdot, \cdot) = Q_{w_t, t}(\cdot, \cdot)$.
7:         Execute $a_t = \operatorname{argmax}_{a_t} \widehat{Q}(s_t, a)$ and receive feedback $s_{t+1}, r_t$.
8:         Update posterior $\beta_t$ with evidence $(a_t, r_t, s_{t+1})$.
9:     **end for**
10: **end for**

---

## A.2 DETAILS OF THE BOORL ALGORITHM

---

**Algorithm 4** BOORL, Offline Phase

---

1: **Require**: Ensemble size $N$, offline dataset $\mathcal{D}^{\text{off}}$, masking distribution $M$
2: Initialize parameters of $N$ independent TD3+BC agents $\{Q_{\theta_i}, \pi_{\phi_i}\}_{i=1}^N$
3: **for** $i = 1, \cdots, N$ **do**
4:     Sample bootstrap mask $m \sim M$
5:     Add $m$ to $\mathcal{D}^{\text{off}}$ as $\mathcal{D}_i^{\text{off}}$
6:     **for** each training iteration **do**
7:         Sample a random minibatch $\{\tau_j\}_{j=1}^B \sim \mathcal{D}_i^{\text{off}}$
8:         Calculate $L_{\text{critic}}^{\text{offline}}(\theta_i)$ and update $\theta_i$
9:         Calculate $L_{\text{actor}}^{\text{offline}}(\phi_i)$ and update $\phi_i$
10:     **end for**
11: **end for**
12: **Return** $\{Q_{\theta_i}, \pi_{\phi_i}\}_{i=1}^N$

---

---

**Algorithm 5** BOORL, Online Phase

---

1: **Require**: $\{Q_{\theta_i}, \pi_{\phi_i}\}_{i=1}^N$, offline dataset $\mathcal{D}^{\text{off}}$
2: Initialize empty online replay buffer $\mathcal{D}^{\text{on}}$
3: **for** each iteration **do**
4:     Obtain initial state from environment $s_0$
5:     **for** step $t = 1, \cdots, T$ **do**
6:         Pick an policy to act $a_t \sim \pi_{\phi_n}(\cdot \mid s_t)$ using $n \sim \text{Uniform}\{1, \cdots, N\}$
7:         Store transition $(s_t, a_t, r_t, s_{t+1})$ in $\mathcal{D}^{\text{on}}$
8:         Sample minibatch $b$ from $\mathcal{D}^{\text{off}}$ and $\mathcal{D}^{\text{on}}$
9:         **for** $i = 1, \cdots, N$ **do**
10:           With $b$, calculate $L_{\text{critic}}^{\text{online}}(\theta_i)$ and update $\theta_i$
11:         **end for**
12:         **for** $i = 1, \cdots, N$ **do**
13:           With $b$, calculate $L_{\text{actor}}^{\text{online}}(\phi_i)$ and update $\phi_i$
14:         **end for**
15:     **end for**
16: **end for**

---

## B   ADDITIONAL PROPOSITIONS

**Proposition B.1.** *Suppose the following equation for each episode holds with $\Gamma_t \leq \Gamma$ for all $k \in [K], h \in [H]$,*

$$\mathbb{E}_k[\Delta_k] \leq \sum_{h=1}^{H} \Gamma_t \sqrt{I_t(w_h; a_t, r_t, s_{t+1})} + \epsilon_t. \tag{14}$$

*Then*

$$\mathbb{E}[Regret(T, \pi)] \leq \Gamma \sqrt{TI(w; \mathcal{H}_T)} + \mathbb{E}\sum_{k=1}^{K}\sum_{h=1}^{H} \epsilon_t.$$

*Proof.* By leveraging the chain rule of mutual information, i.e.

$$I(X; Y_1, \ldots, Y_N) = \sum_{i=1}^{N} I(X; Y_i | Y_1, \ldots, Y_{i-1}),$$

and the Cauchy-Schwartz inequality, we have the result immediately. ∎

## C   MISSING PROOFS

### C.1   PROOF OF THEOREM 3.1

*Proof.* By regret decomposition in Lemma D.1, we have

$$\mathbb{E}_k[\Delta_k] = \sum_{h=1}^{H} \mathbb{E}_{\pi^*}\left[\langle Q_t(s_h, \cdot), \pi_h^*(\cdot \mid s_h) - \pi_{h,k}(\cdot \mid s_h)\rangle\right] + \sum_{h=1}^{H}\left(\mathbb{E}_{\pi^*}[\iota_t(s, a)] - \mathbb{E}_k[\iota_t(s, a)]\right)$$

$$\leq \sum_{h=1}^{H}\left(\mathbb{E}_{\pi^*}[\iota_t(s, a)] - \mathbb{E}_k[\iota_t(s, a)]\right),$$

where $\iota_t(s, a) = r_t(s, a) + (\mathbb{B}_h V_{t+1})(s, a) - Q_{t+1}(s, a)$. The inequality is due to the fact that $\pi_t$ is the greedy policy with respect to $Q_t$.

Let $\mathcal{W}_k$ be the confidence set of $w$ such that Equation (6) holds, we have

$$\mathbb{P}_k(w \in \mathcal{W}_k) \geq 1 - \frac{\delta}{2}.$$

For a UCB algorithm with upper confidence estimation $Q_t(s, a) = \bar{Q}_{t,w}(s, a) + \frac{\Gamma_t}{2}\sqrt{I_t(w_h; r_{t,a}, s_{t+1,a})}$, we have

$$-\Gamma_t\sqrt{I_t(w_h; r_{t,a}, s_{t+1,a})} \leq \iota_t(s, a) \leq 0$$

when $w \in \mathcal{W}_k$.Then we have

$$\sum_{h=1}^{H} (\mathbb{E}_{\pi^*}[\iota_t(s,a)] - \mathbb{E}_k[\iota_t(s,a)])$$

$$\leq \sum_{h=1}^{H} \left\{ \mathbb{1}_{w \in \mathcal{W}_k} \{\mathbb{E}_{\pi^*}[\iota_t(s,a)] - \mathbb{E}_k[\iota_t(s,a)]\} + \frac{1}{2}\delta \cdot 2H \right\}$$

$$\leq \sum_{h=1}^{H} \mathbb{E}_k \left[ \sum_{a \in \mathcal{A}} \mathbb{P}(a_t = a)\Gamma_t \sqrt{I_t(w_h; r_{t,a}, s_{t+1,a})} \right] + \delta H^2$$

$$\leq \sum_{h=1}^{H} \mathbb{E}_k \left[ \Gamma_t \sqrt{\sum_{a \in \mathcal{A}} \mathbb{P}(a_t = a)I_t(w_h; r_{t,a}, s_{t+1,a})} \right] + \delta H^2$$

$$= \sum_{h=1}^{H} \mathbb{E}_k \left[ \Gamma_t \sqrt{\sum_{a \in \mathcal{A}} \mathbb{P}(a_t = a)I_t(w_h; r_{t,a_t}, s_{t+1,a_t} \,|\, a_t = a)} \right] + \delta H^2$$

$$= \sum_{h=1}^{H} \mathbb{E}_k \left[ \Gamma_t \sqrt{I_t(w_h; a_t, r_t, s_{t+1})} \right] + \delta H^2.$$

Similarly, for a LCB algorithm with lower confidence functions $Q_t(s,a) = \bar{Q}_{t,w}(s,a) - \frac{\Gamma_t}{2}\sqrt{I_t(w_h; r_{t,a}, s_{t+1,a})}$, we have

$$0 \leq \iota_t(s,a) \leq \Gamma_t \sqrt{I_t(w_h; r_{t,a}, s_{t+1,a})}$$

when $w \in \mathcal{W}_k$.Then we have

$$\sum_{h=1}^{H} (\mathbb{E}_{\pi^*}[\iota_t(s,a)] - \mathbb{E}_k[\iota_t(s,a)])$$

$$\leq \sum_{h=1}^{H} \left\{ \mathbb{1}_{w \in \mathcal{W}_k} \{\mathbb{E}_{\pi^*}[\iota_t(s,a)] - \mathbb{E}_k[\iota_t(s,a)]\} + \frac{1}{2}\delta \cdot 2H \right\}$$

$$\leq \sum_{h=1}^{H} \mathbb{E}_{\pi^*} \left[ \sum_{a \in \mathcal{A}} \mathbb{P}(a_t^* = a)\Gamma_t \sqrt{I_t(w_h; r_{t,a}, s_{t+1,a})} \right] + \delta H^2$$

$$\leq \sum_{h=1}^{H} \mathbb{E}_{\pi^*} \left[ \Gamma_t \sqrt{\sum_{a \in \mathcal{A}} \mathbb{P}(a_t^* = a)I_t(w_h; r_{t,a}, s_{t+1,a})} \right] + \delta H^2$$

$$= \sum_{h=1}^{H} \mathbb{E}_{\pi^*} \left[ \Gamma_t \sqrt{\sum_{a \in \mathcal{A}} \mathbb{P}(a_t^* = a)I_t(w_h; r_{t,a_t^*}, s_{t+1,a_t^*} \,|\, a_t^* = a)} \right] + \delta H^2$$

$$= \sum_{h=1}^{H} \mathbb{E}_{\pi^*} \left[ \Gamma_t \sqrt{I_t(w_h; a_t^*, r_t, s_{t+1})} \right] + \delta H^2.$$

For Thompson Sampling, note that the probability matching property implies that $\mathbb{P}_k(\widehat{w}_k \in \mathcal{W}_k) = \mathbb{P}_k(w \in \mathcal{W}_k) \geq 1 - \frac{\delta}{2}$, we have

$$\sum_{h=1}^{H} \left( \mathbb{E}_{\pi^*}[\iota_t(s,a)] - \mathbb{E}_k[\iota_t(s,a)] \right)$$

$$\leq \sum_{h=1}^{H} \left\{ \mathbb{1}_{w,\widehat{w}_k \in \mathcal{W}_k} \left\{ \mathbb{E}_{\pi^*}[\iota_t(s,a)] - \mathbb{E}_k[\iota_t(s,a)] \right\} + \delta \cdot 2H \right\}$$

$$\leq \sum_{h=1}^{H} \mathbb{E}_{\pi^*} \left[ \sum_{a \in \mathcal{A}} \mathbb{P}(a_t^* = a) \Gamma_t \sqrt{I_t(w_h; r_{t,a}, s_{t+1,a})} \right] + 2\delta H^2$$

$$= \sum_{h=1}^{H} \mathbb{E}_k \left[ \sum_{a \in \mathcal{A}} \mathbb{P}(a_t = a) \Gamma_t \sqrt{I_t(w_h; r_{t,a}, s_{t+1,a})} \right] + 2\delta H^2.$$

The last equality and the last equality is due to the fact that $\pi_{k,h}$ is the optimal policy under $\widehat{w}_k$, and $w$ and $\widehat{w}_k$ have the same distribution. The rest of the proof is similar to the case of UCB and LCB and is omitted for simplicity. $\qquad\square$

### C.2 PROOF OF THEOREM 3.2

**Theorem C.1** (Regret of Bayesian Agents in Linear MDPs, restatement). *Given an offline dataset $\mathcal{D}$ of size $N$, and a fixed posterior $\beta$ during the online interaction phase, the regret of Thompson sampling during online interaction satisfies the following bound:*

$$\mathrm{BayesRegret}(N, T, \pi) \leq 4c\sqrt{d^3 H^3 \iota} \left( \sqrt{\frac{N}{C_\beta^\dagger} + T} - \sqrt{\frac{N}{C_\beta^\dagger}} \right), \tag{15}$$

*for sufficiently large $N$ or $T$. Here $\iota$ is a logarithmic factor and $c$ is an absolute constant.*

*Proof.* At each online episode $k$, we have

$$\sum_{h=1}^{H} \Gamma_t \mathbb{E}_{\pi^*} \left[ \sqrt{I_t(w_h; a_t^*, r_t, s_{t+1})} \right]$$

$$= \sum_{h=1}^{H} \Gamma_t \mathbb{E}_{\pi^*} \left[ \log \left( 1 + \phi(s_h, a_h)^\top \Lambda_h^{-1} \phi(s_h, a_h) \right)^{1/2} \right]$$

$$\leq \Gamma_t \mathbb{E}_{\pi^*} \left[ \sum_{h=1}^{H} \left( \phi(s_t, a_t)^\top \Lambda_k^{-1} \phi(s_t, a_t) \right)^{1/2} \right]$$

$$= \Gamma_t \mathbb{E}_{\pi^*} \left[ \sum_{h=1}^{H} \sqrt{\mathrm{Tr} \left( \phi(s,a) \phi(s,a)^\top \Lambda_k^{-1} \right)} \right]$$

$$\leq \Gamma_t \sum_{h=1}^{H} \sqrt{\mathrm{Tr} \left( \mathbb{E}_{d_h^{\pi^*}} \left[ \phi(s,a) \phi(s,a)^\top \right] \Lambda_t^{-1} \right)}$$

$$= \Gamma_t \sum_{h=1}^{H} \sqrt{\mathrm{Tr} \left( \Sigma_{\pi^*,h}^\top \Lambda_t^{-1} \right)}, \tag{16}$$

where $\Sigma_{\pi^*,h} \triangleq \mathbb{E}_{d_h^{\pi^*}} \left[ \phi(s,a) \phi(s,a)^\top \right]$. The first equality uses Lemma D.3, The first inequality uses the fact that $\log(1 + x) \leq x \, \forall x \geq 0$. The second equality uses the trace trick and the last inequality due to Jensen inequality and the linearity of the trace function.

By the definition of Bayesian coverage coefficient, we have

$$\mathbb{E} \left[ \sum_{\ell=1}^{L} \phi(s_{\ell,h}, a_{\ell,h}) \phi(s_{\ell,h}, a_{\ell,h})^\top \right] \succeq \frac{L}{C_\beta^\dagger} \Sigma_{\pi_\beta^*, h},$$

where $\Sigma_{\pi^*_\beta, h} \triangleq \mathbb{E}_{w \sim \beta} \mathbb{E}_{d_h^{\pi^*_w}} [\phi(s,a)\phi(s,a)^\top]$.

From the probability matching property of Thompson sampling method, we have

$$\mathbb{E}\left[\sum_{k=1}^K \phi(s_t, a_t)\phi(s_t, a_t)^\top\right] = K\Sigma_{\pi^*_\beta, h}.$$

From matrix concentration inequalities (Gittens & Tropp, 2011), with a probability $1 - \xi$ where $\xi = \frac{d}{H}e^{-\frac{4(L+KC^\dagger_\beta)}{C^\dagger_\beta \kappa^2_\beta}}$ , we have

$$\sum_{\ell=1}^L \phi(s_{\ell,h}, a_{\ell,h})\phi(s_{\ell,h}, a_{\ell,h})^\top + \sum_{k=1}^K \phi(s_t, a_t)\phi(s_t, a_t)^\top \succeq \frac{1}{2}\left(\frac{L}{C^\dagger_\beta} + K\right)\Sigma_{\pi^*_\beta}. \tag{17}$$

Here $\kappa_\beta = \max_{h \in [H]} \frac{\lambda_{\max}(\Sigma_{\pi^*_\beta, h})}{\lambda_{\min}(\Sigma_{\pi^*_\beta, h})}$ is the condition number for the feature matrix under expert policy $\pi^*$, $\lambda_{\max}$ is the largest eigenvalue and $\lambda_{\min}$ is the smallest *non-zero* eigenvalue. Let $\mathcal{E}$ be the event such that Equation (17) holds, then we have

$$\mathbb{E}_\beta[\Delta_k]$$
$$\leq \mathbb{E}_\beta\left[\sum_{h=1}^H \Gamma_t \mathbb{E}_{\pi^*}\left[\sqrt{I_t(w_h; a^*_t, r_t, s_{t+1})}\right]\right] + 2\delta H^2$$
$$\leq \Gamma \sum_{h=1}^H \sqrt{\mathbb{E}_\beta \operatorname{Tr}\left(\Sigma^\top_{\pi^*, h}\Lambda^{-1}_t\right)} + 2\delta H^2$$
$$\leq \mathbb{1}_\mathcal{E}\left\{\Gamma \sum_{h=1}^H \sqrt{\mathbb{E}_\beta \operatorname{Tr}\left(\Sigma^\top_{\pi^*, h}\Lambda^{-1}_t\right)}\right\} + \xi H^2 + 2\delta H^2$$
$$\leq \Gamma \sum_{h=1}^H \sqrt{\mathbb{E}_\beta \operatorname{Tr}\left(\Sigma^\top_{\pi^*, h}\left(\lambda \cdot I + \frac{1}{2}\left(\frac{L}{C^\dagger_\beta} + K\right)\Sigma_{\pi^*, h}\right)^{-1}\right)} + \xi H^2 + 2\delta H^2$$
$$\leq \Gamma \sum_{h=1}^H \sqrt{\sum_{j=1}^d \frac{\lambda_j(h)}{\lambda + \frac{1}{2}\left(\frac{L}{C^\dagger_\beta} + K\right) \cdot \lambda_j(h)}} + (2\delta + \xi)H^2.$$

Here $\{\lambda_j(h)\}_{j=1}^d$ are the eigenvalues of $\Sigma_{\pi^*, h}$ for all $h \in [H]$. The first inequality follows from Lemma D.4, and the second to last inequality follows from the Jensen inequality and the definition of event $\mathcal{E}$.

Meanwhile, by definition, we have $\|\phi(s,a)\| \leq 1$ for all $(s,a) \in \mathcal{S} \times \mathcal{A}$. By Jensen's inequality, we have

$$\|\Sigma_{\pi^*, h}\|_{\mathrm{op}} \leq \mathbb{E}_{\pi^*}\left[\|\phi(s,a)\phi(s,a)^\top\|_{\mathrm{op}}\right] \leq 1 \tag{18}$$

for all $h \in [H]$. As $\Sigma_{\pi^*, h}$ is positive semidefinite, we have $\lambda_j(h) \in [0, 1]$ for all $h \in [H]$ and all $j \in [d]$.

Then we have

$$
\begin{aligned}
\mathbb{E}_\beta[\Delta_k] \leq & \Gamma \sum_{h=1}^{H} \sqrt{\sum_{j=1}^{d} \frac{1}{\lambda + \frac{1}{2}(\frac{L}{C_\beta^\dagger} + K)}} + (2\delta + \xi)H^2 \\
\leq & H\Gamma \sqrt{\frac{2d}{\frac{L}{C_\beta^\dagger} + K}} + (2\delta + \xi)H^2 \\
\leq & 2c\sqrt{\frac{2d^3 H^3 \iota}{\frac{L}{C_\beta^\dagger} + K}} + (2\delta + \xi)H^2,
\end{aligned}
$$

where $\iota = \log \frac{4dT}{\delta}$. For sufficiently large $L$ and $K$ such that $\xi = (\frac{L}{C_\beta^\dagger} + K)^{-1/2} K^{-1}$ and let $\delta = (\frac{L}{C_\beta^\dagger} + K)^{-1/2} K^{-1}$. Using the fact that

$$
\sum_{k=1}^{K} \sqrt{\frac{1}{a+bk}} \leq \int_0^K \sqrt{\frac{1}{a+bx}} dx \leq \frac{2}{b}(\sqrt{a+bK} - \sqrt{a}),
$$

we have the desired result. $\hfill\square$

### C.3 PROOF OF PROPOSITION 3.3

*Proof.* Let $T = 1$ in Theorem 3.2 and note that $\sqrt{x+1} - \sqrt{x} \leq 2/\sqrt{x}$, we have the result immediately. $\hfill\square$

### C.4 PROOF OF PROPOSITION 3.4

*Proof.* Let $N = 0$ in Theorem 3.2, and we have the result immediately. $\hfill\square$

## D  AUXILIARY LEMMAS

**Lemma D.1** (Regret Decomposition (Cai et al., 2020)). *We define the model prediction error as*
$$
\iota_{k,h}(s,a) = r_{k,h}(s,a) + (\mathbb{B}_h V_{k,h+1})(s,a) - Q_{k,h+1}(s,a), \tag{19}
$$
*which arises from estimating $\mathbb{P}_h V_{h+1}^k$ in the Bellman equation based on only finite historical data. Also, we define the following filtration generated by the state-action sequence and reward functions.*

**Definition D.2** (Filtration). For any $(t) \in [K] \times [H]$, we define $\mathcal{F}_{t,1}$ as the $\sigma$-algebra generated by the following state-action sequence and reward functions,
$$
\{(s_{\tau,i}, a_{\tau,i})\}_{(\tau,i)\in[k-1]\times[H]} \cup \{r^\tau\}_{\tau\in[k]} \cup \{(s_{k,h}, a_{k,h})\}_{i\in[h]},
$$
and $\mathcal{F}_{t,2}$ as the $\sigma$-algebra generated by
$$
\{(s_{\tau,i}, a_{\tau,i})\}_{(\tau,i)\in[k-1]\times[H]} \cup \{r^\tau\}_{\tau\in[k]} \cup \{(s_{k,h}, a_{k,h})\}_{i\in[h]} \cup \{s_{h+1}^k\},
$$
where, for the simplicity of discussion, we define $s_{H+1}^k$ as a null state for any $k \in [K]$.

*It holds that*

$$
\begin{aligned}
Regret(T) = & \sum_{k=1}^{K} \big(V_1^{\pi^*,k}(s_1^k) - V_1^{\pi^k,k}(s_1^k)\big) \\
= & \sum_{k=1}^{K} \sum_{h=1}^{H} \mathbb{E}_{\pi^*}\big[\langle Q_h^k(s_h, \cdot), \pi_h^*(\cdot \mid s_h) - \pi_{k,h}(\cdot \mid s_h)\rangle\big] + \mathcal{M}_{K,H,2} \\
& + \sum_{k=1}^{K} \sum_{h=1}^{H} \big(\mathbb{E}_{\pi^*}[\iota_h^k(s_h, a_h)] - \iota_h^k(s_{k,h}, a_{k,h})\big).
\end{aligned} \tag{20}
$$

*Here $\{\mathcal{M}_{t,m}\}_{(t,m)\in[K]\times[H]\times[2]}$ is a martingale adapted to the filtration $\{\mathcal{F}_{t,m}\}_{(t,m)\in[K]\times[H]\times[2]}$.*

*Proof.* See Lemma 4.2 in Cai et al. (2020) for a detailed proof. □

**Lemma D.3** (Mutual Information in Linear MDP). *It hold that*

$$I_t(w_h; a_t, r_t, s_{t+1}|\mathcal{D}) = \frac{1}{2} \log\left(1 + \phi(s_t, a_t)^\top \Lambda_t^{-1} \phi(s_t, a_t)\right).$$

*Proof.* Let the prior be $w_h \sim \mathcal{N}(0, \lambda \cdot I)$, then we have the following closed form posterior

$$w_h | \mathcal{D} \sim \mathcal{N}(\widehat{w}_h, \Lambda_t^{-1}),$$

where

$$\widehat{w}_h = \Lambda_h^{-1} \left( \sum_{k=1}^{K} \phi(s_t, a_t) \cdot (r_t + \widehat{V}_{h+1}(s_{t+1})) \right),$$

$$\Lambda_h = \sum_{k=1}^{K} \phi(s_t, a_t) \phi(s_t, a_t)^\top + \lambda \cdot I.$$

Note that this is equivalent to the regularized least-square solution for linear MDPs (Jin et al., 2021). Then we have

$$\begin{aligned}
I_t(w_h; a_t, r_t, s_{t+1}|\mathcal{D}) &= H(w_h|\mathcal{D}) - H(w_h|\mathcal{D} \cup \{(r_t, a_t, s_{t+1})\}) \\
&= \frac{1}{2} \log \frac{\det(\Lambda_t^\dagger)}{\det(\Lambda_t)} \\
&= \frac{1}{2} \log \det(I + \Lambda_t^{-1/2} \phi(s_t, a_t) \phi(s_t, a_t)^\top \Lambda_t^{-1/2}) \\
&= \frac{1}{2} \log\left(1 + \phi(s_t, a_t)^\top \Lambda_t^{-1} \phi(s_t, a_t)\right).
\end{aligned}$$

where $\Lambda_t^\dagger = \Lambda_t + \phi(s_h, a_h) \phi(s_h, a_h)^\top$. □

**Lemma D.4.** *Under linear MDP, we have*

$$\mathbb{P}_k\left(\left|Q_{t,w}(s, a) - \bar{Q}_{t,w}(s, a)\right| \le \frac{\Gamma_t}{2} \sqrt{I_t(w_h; r_{t,a}, s_{t+1,a})}, \forall h \in [H], s \in \mathcal{S}, a \in \mathcal{A}\right) \ge 1 - \frac{\delta}{2}$$

*With $\Gamma_t \equiv \Gamma = 2cHd\sqrt{\log \frac{4dT}{\delta}}$, where c is an absolute constant and $\bar{Q}_{t,w}(s, a) = r_{h,w}(s, a) + \mathbb{P}_h V_{h+1,w}(s, a)$.*

*Proof.* Following a similar argument in Lemma 5.2 in Jin et al. (2021), We have with probability $1 - \delta$,

$$\begin{aligned}
&\left|Q_{t,w}(s, a) - \bar{Q}_{t,w}(s, a)\right| \\
\le& \beta \sqrt{\phi(s, a) \Lambda_h^{-1} \phi(s, a)} \\
\le& \beta \sqrt{\log(1 + \phi(s, a)\Lambda_h^{-1}\phi(s, a)) \cdot \frac{\phi(s, a)\Lambda_h^{-1}\phi(s, a)}{\log(1 + \phi(s, a)\Lambda_h^{-1}\phi(s, a))}} \\
\le& \beta \sqrt{2\log(1 + \phi(s, a)\Lambda_h^{-1}\phi(s, a))} \\
\le& 2\beta \sqrt{I_t(w_h; r_{t,a}, s_{t+1,a})},
\end{aligned}$$

where $\beta = cHd\sqrt{\log \frac{4dT}{\delta}}$ and $\Lambda_h$ is defined as in Lemma D.3. The last inequality use the fact that $\phi(s, a)\Lambda_h^{-1}\phi(s, a) \le 1$ and $2\log(1 + x) \ge x$ for $x \in [0, 1]$. The last step follows from Lemma D.3.

□

# E   EXPERIMENTS ON MULTI-ARM BANDITS

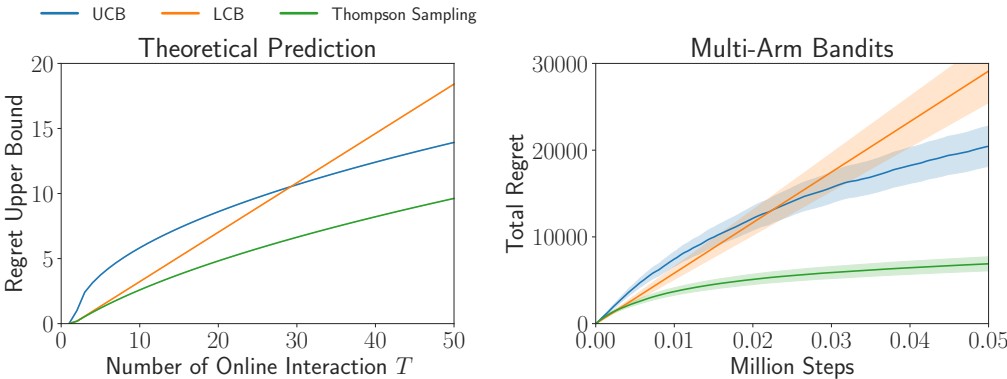

Figure 4: Theoretical prediction of Theorem 3.2 and experiments on multi-arm bandits.

We conducted experiments on a multi-arm bandit model with 10 arms, where each arm pull yielded a reward based on probability $p_i$. The ultimate objective was to identify the arm associated with the maximum probability, $p^* = \max\{p_i\}_{i=1}^n$. Data collection was facilitated by using a random policy to pull the arms, generating an offline dataset. Statistical attributes calculated from this dataset were employed to derive the Upper Confidence Bound (UCB) policy, $\pi_{\text{UCB}}$; the Lower Confidence Bound (LCB) policy, $\pi_{\text{LCB}}$; and the Thompson Sampling (TS) policy, $\pi_{\text{TS}}$.

Subsequent arm pulls leveraged these policies, updating statistical variables each iteration. The regret, defined as the difference between $p^*$ and the chosen policy's probability $p_\pi$, was computed in each round. Accumulative regret was plotted and illustrated in Figure 4. The results confirmed the effective alignment of the experimental outcome with the theoretical results expressed by Equation 15 in the context of the multi-arm bandit model.

Specifically, in the bandit setting, we let each arm $a$ has a probability $\theta$ to yield a reward of 1, and a probability $1 - \theta$ to yield a reward of 0. the parameters $\theta$ are i.i.d. drawn from a Beta distribution for all arms $a_n$. We use 10 arms, 1000 offline data points and the online phase lasts for 100000 steps.

# F COMPLETE EXPERIMENTAL RESULTS

The results in Figure 6 show TD3+BC exhibits safe but slow performance improvement, resulting in worse asymptotic performance. On the other hand, TD3 suffers from initial performance degradation, especially in narrow distribution datasets (e.g., expert datasets). Differently, BOORL attains a fast performance improvement with a smaller regret. Due to the offline bootstrap, the initial performance in the online phase between BOORL and baselines exits a small difference, while it does not change the conclusion.

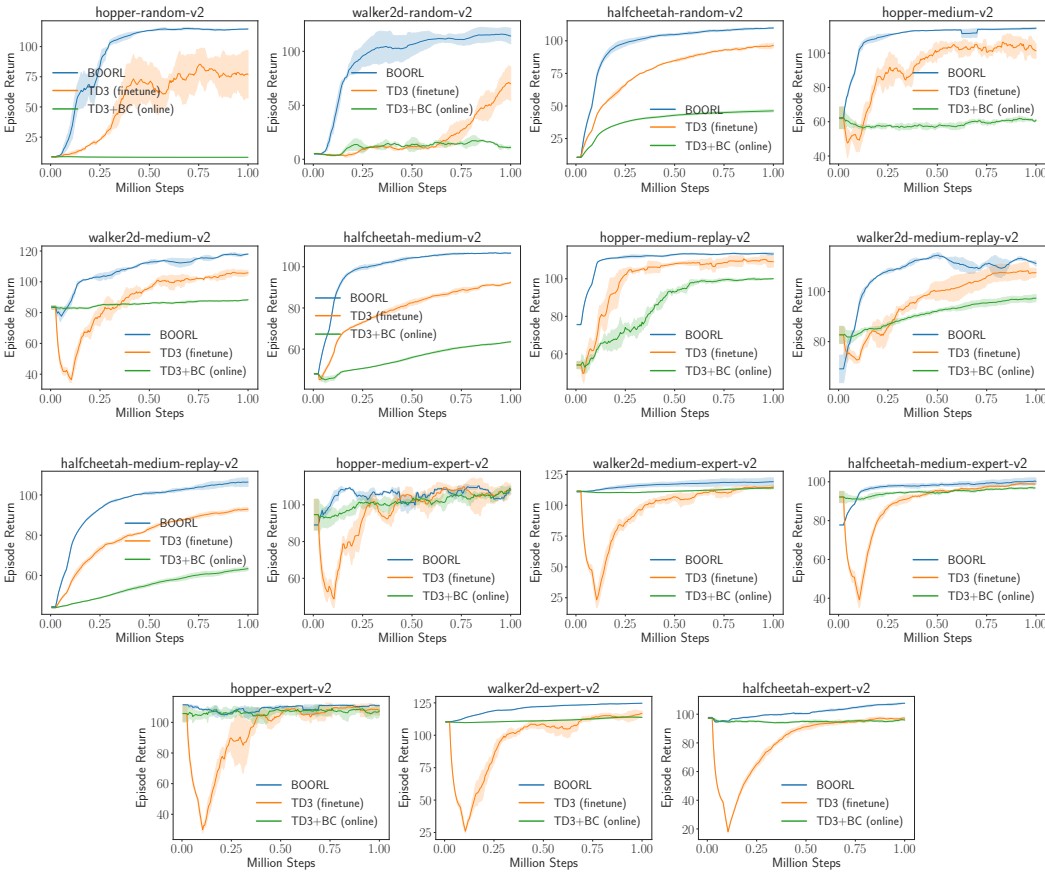

Figure 5: Comparison between BOORL and baselines in the finetune phase. We adopt datasets of various quality for offline training and then load same pre-trained weight for online learning. We adopt normalized score metric averaged with five random seeds.

# G  COMPARISON WITH PEX

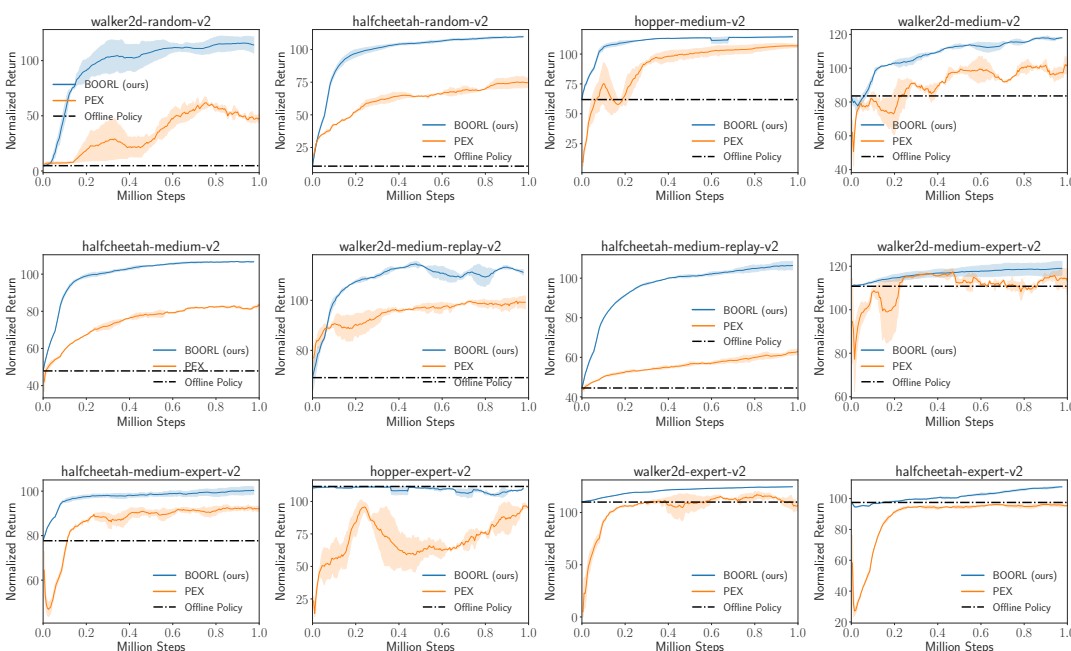

Figure 6: Comparison between BOORL and PEX in the finetune phase. We adopt datasets of various quality for offline training and then load same pre-trained weight for online learning. We adopt normalized score metric averaged with five random seeds.

# H  ADDITIONAL EXPERIMENTS

## H.1  ABLATION FOR BOORL

We aim to understand the behavior of BOORL by performing ablation studies. (1) We store the offline and online data into the same buffer for uniform sampling, named BOORL (Uniform Buffer). (2) We set the Ensemble Number to 1 to investigate the effect of the Thompson Sampling, named BOORL (Ensemble Num=1). (3) Similarly, we set UTD ratio $G$ to 1 to investigate the effect of UTD, named BOORL (UTD=1). The experimental results in Figure 7 show that each module is essential to the superior performance of our algorithm. In the ablation studies, we use TD3+BC and TD3 as the backbone of offline and online algorithms.

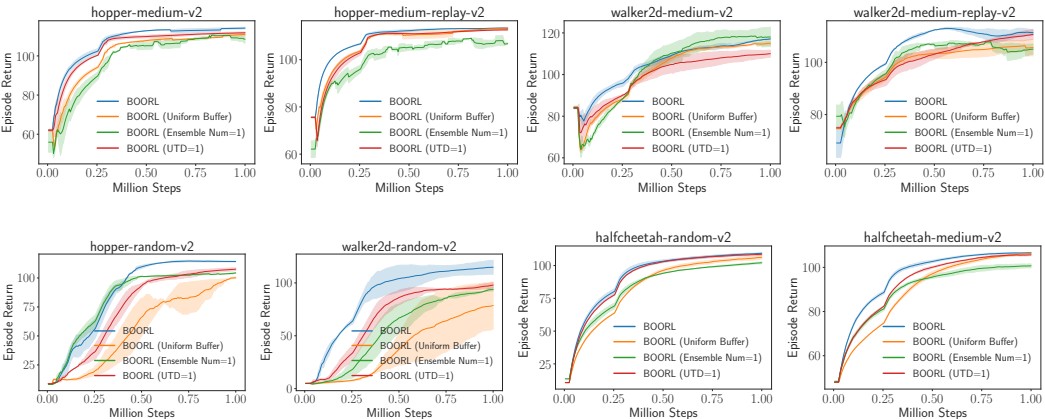

Figure 7: Module ablation study of BOORL.

## H.2  ABLATION FOR MASK RATIO

In addition, we conduct ablation studies for mask ratio $p$. The experimental results in Figure 8 show that the performance of our algorithm is robust to the changes of $p$. Similar results are also found in Osband et al. (2016). Therefore, we select the uniform parameter $p = 0.9$ in all experiments.

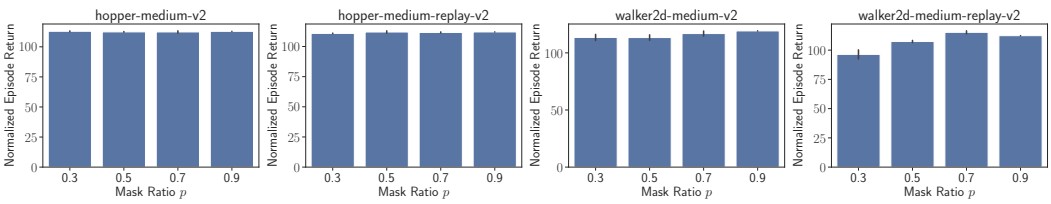

Figure 8: The performance comparison between various mask ratios $p$.

## H.3  ABLATION FOR COMPUTATIONAL OVERHEAD

We aim to provide a comparison between the computational overhead of using an ensemble versus not using an ensemble in our approach. Specifically, we train our method with various ensemble sizes N on the same computational device (GeForce RTX 3090 GPU). The time required to complete 1M training is shown in Table 4. Since we adopt the multi-head structure and share part of the network, the computational overhead does not increase significantly as the number of N increases.

| Ensemble Size | 1 | 5 | 10 | 20 |
|---|---|---|---|---|
| Computational Overhead | 2.5 h | 2.7 h | 2.9 h | 3.2 h |

Table 4: Ablation results for computational overhead.

# I ADDITIONAL IMPLEMENTATION SPECS

We aim to make minimal modifications to existing offline algorithms while achieving maximum performance improvement. The following method is general in the offline-to-online setting. As for different offline algorithms, we just need to substitute corresponding critic and actor update loss.

**Offline Phase** We first randomly initialize $N$ policy networks and corresponding $Q$-value networks $\{\pi_{\phi_i}, Q_{\theta_i}\}_{i=1}^N$. Similar with Osband et al. (2016), we use the Bernoulli mask $m_1, m_2, \cdots, m_N \in$ Ber $(p)$ for each policy to implement an offline bootstrap, where $p = 0.9$. These flags are stored in the memory replay buffer $\mathcal{D}_i^{\text{off}}$ and identify which policies are trained on which data. Next, each one of these offline policies is trained against its own pessimistic $Q$-value network and bootstrapped dataset with the offline RL loss (e.g., TD3+BC (Fujimoto & Gu, 2021)):

$$\mathcal{L}_{\text{critic}}(\theta_i) = \mathbb{E}_{(s,a,r,s',m_i)\sim\mathcal{D}_i^{\text{off}}} \left[ \left( r + \gamma Q_{\theta_i'}(s',\widetilde{a}) - Q_{\theta_i}(s,a) \right)^2 * m_i \right] \tag{21}$$

$$\mathcal{L}_{\text{actor}}(\phi_i) = -\mathbb{E}_{(s,a,m_i)\sim\mathcal{D}_i^{\text{off}}} \left[ \left( \lambda Q_{\theta_i}(s, \pi_{\phi_i}(s)) - (\pi_{\phi_i}(s) - a)^2 \right) * m_i \right], \tag{22}$$

where $\widetilde{a} = \pi_{\phi_i'}(s') + \epsilon$ is the target plicy smoothing regularization and $\lambda$ is the hyper-parameter for behavior cloning.

**Online Phase** We first load the offline pre-trained model. Then, we approximate a bootstrap sample by selecting $n \in \{1, \cdots, N\}$ uniformly at random at each time step and following $\pi_n$ to collect online data. Each loaded policy and $Q$-value network is continued to be trained with the online RL loss (e.g., TD3 (Fujimoto et al., 2018)):

$$\mathcal{L}_{\text{critic}}(\theta_i) = \mathbb{E}_{(s,a,r,s')\sim\mathcal{D}^{\text{off}}\cup\mathcal{D}^{\text{on}}} \left[ \left( r + \gamma Q_{\theta_i'}(s',\widetilde{a}) - Q_{\theta_i}(s,a) \right) \right]^2 \tag{23}$$

$$\mathcal{L}_{\text{actor}}(\phi_i) = -\mathbb{E}_{(s,a)\sim\mathcal{D}^{\text{off}}\cup\mathcal{D}^{\text{on}}} \left[ Q_{\theta_i}(s, \pi_{\phi_i}(s)) \right], \tag{24}$$

As for the sample selection, we adopt a simple yet efficient sample selection method (Ball et al., 2023) to incorporate prior data better. For each batch, we sample $50\%$ of the data from the online replay buffer $\mathcal{D}^{\text{on}}$, and the remaining $50\%$ from the offline replay buffer $\mathcal{D}^{\text{off}}$. Further, we increase the UTD ratio $G$ to make the Bellman backups perform as sample-efficiently as possible, where $G = 5$.

## J    EXPERIMENTAL DETAILS

**Experimental Setting.**    For TD3+BC (online), TD3 (finetune), and IQL (online), we first load the offline dataset into the online replay buffer and add the online collected data into the buffer. Then, we uniformly sample data to train from the online buffer.

**Hyper-parameters.**    We adopt the TD3+BC and TD3 as the backbone of offline and online algorithms. Therefore, we build BOORL based on the code of the TD3+BC. We outline the hyper-parameters used by BOORL in Table 5.

| Hyperparameter | Value |
|---|---|
| Optimizer | Adam |
| Critic learning rate | 3e-4 |
| Actor learning rate | 3e-4 |
| Mini-batch size | 256 |
| Discount factor | 0.99 |
| Target update rate | 5e-3 |
| Policy noise | 0.2 |
| Policy noise clipping | (-0.5, 0.5) |
| TD3+BC parameter $\alpha$ | 2.5 |
| IQL parameter $\tau$ | 0.9 |
| Architecture | Value |
| Critic hidden dim | 256 |
| Critic hidden layers | 2 |
| Critic activation function | ReLU |
| Actor hidden dim | 256 |
| Actor hidden layers | 2 |
| Actor activation function | ReLU |
| BOORL Parameters | Value |
| Mask ratio $p$ | 0.9 |
| Ensemble Number | 5 |
| UTD ratio $G$ | 5 |

Table 5: Hyper-parameters sheet of BOORL.

**Baselines Implementation.**    We adopt the author-provided implementations from GitHub for TD3 [1], TD3+BC [2], CQL [3], IQL [4], Off2On [5], ODT [6], PEX [7] and Cal-QL [8]. We use the official implementation in the author-provided code for TD3+BC (online) and IQL (online). All experiments are conducted on the same experimental setup, a single GeForce RTX 3090 GPU and an Intel Core i7-6700k CPU at 4.00GHz.

---

[1] https://github.com/sfujim/TD3
[2] https://github.com/sfujim/TD3_BC
[3] https://github.com/aviralkumar2907/CQL
[4] https://github.com/ikostrikov/implicit_q_learning
[5] https://github.com/shlee94/Off2OnRL
[6] https://github.com/facebookresearch/online-dt
[7] https://github.com/Haichao-Zhang/PEX
[8] https://github.com/nakamotoo/Cal-QL

