# OpenReview forum: "Bayesian Offline-to-Online Reinforcement Learning : A Realist Approach"
_ICLR.cc/2024/Conference — Submitted to ICLR 2024_

### Official Review · Reviewer_JwaT · 2023-10-28

**Soundness:** 3 good
**Presentation:** 3 good
**Contribution:** 3 good
**Rating:** 6
**Confidence:** 4

**Summary:**

This paper addresses the challenge of fine-tuning pre-trained offline Reinforcement Learning (RL) agents. Specifically, the study introduces a Bayesian approach called BOORL, where the dataset is treated as priors and online interactions are utilized to update posteriors. By sampling actions from these posteriors, this method effectively avoids excessive optimism or pessimism in off-to-on settings. Experimental results on the D4RL benchmark demonstrate that BOORL outperforms other baseline methods.

**Strengths:**

- This paper provides a theoretical analysis in linear MDPs under the offline-to-online settings.
- The motivation and main idea of the proposed method are reasonable and interesting.
- This writing is clear and easy to follow.
- The proposed method outperforms previous baselines in the experiments.

**Weaknesses:**

- Experiments were solely performed on the less challenging locomotion tasks. Obtaining results from the more demanding antmaze tasks would provide stronger evidence.
- The performance of PEX significantly deviates from the original results and my personal experience, indicating a potential bug in the code or suboptimal parameter tuning.
- The legend in Figure 2 can be put to the top of two images to avoid overlapping with the curves.

**Questions:**

- There are some missing SOTA baselines for offline-to-online fine-tuning in the experiments: Reincarnating RL [1] and InAC [2]. Moreover, the current results of PEX seems to be problematic.

- In Figure 5, the three curves usually have the same starting points except for "hopper-medium-replay-v2", "walker-medium-replay-v2", and "halfcheetah-medium-expert-v2". Why does BOORL have a different value at step 0 in these tasks?

[1] (Agarwal et al., NeurIPS' 22) Reincarnating reinforcement learning: Reusing prior computation to accelerate progress

[2] (Xiao et al., ICLR' 23) The In-Sample Softmax for Offline Reinforcement Learning

---

> ### Author Response · Authors · 2023-11-20
> **Response to Reviewer JwaT (Part 1/2)**
>
> Dear Reviewer,
>
> We appreciate that the Reviewer found our work reasonable, clear, and easy to follow. In our response, we clarify the points the Reviewer raised with additional experiments on Antmaze and more baselines.
>
> **W1: Obtaining results from the more demanding Antmaze tasks.**
>
> **A for W1:**
> The results of Antmaze tasks are shown in Table 2 in the original paper. As suggested, we conducted more experiments on the Antmaze tasks.
> The following table shows that BOORL outperforms various baselines on these challenging tasks.
>
>
> | | BOORL | ODT | Cql-QL | PEX |
> | :---: | :---: | :---: | :---: | :---: |
> | antmaze-umaze-v2 | 80.0$\rightarrow$**100.0** | 55.0$\rightarrow$85.0 | 80.0$\rightarrow$**100.0** | 80.0$\rightarrow$95.0 |
> | antmaze-umaze-diverse-v2 | 75.0$\rightarrow$**95.0** | 50.0$\rightarrow$80.0 | 75.0$\rightarrow$**95.0** | 75.0$\rightarrow$80.0 |
> | antmaze-medium-play-v2 | 50.0$\rightarrow$**100.0** | 0.0$\rightarrow$0.0 |  60.0$\rightarrow$90.0 | 70.0$\rightarrow$85.0 |
> | antmaze-medium-diverse-v2 | 60.0$\rightarrow$**85.0** | 0.0$\rightarrow$0.0 | 70.0$\rightarrow$**85.0** | 60.0$\rightarrow$**85.0** |
> | antmaze-large-play-v2 | 60.0$\rightarrow$**75.0** | 0.0$\rightarrow$0.0 | 25.0$\rightarrow$55.0 | 50.0$\rightarrow$55.0|
> | antmaze-large-diverse-v2 | 50.0$\rightarrow$**80.0** | 0.0$\rightarrow$0.0 | 35.0$\rightarrow$70.0 | 45.0$\rightarrow$55.0|
>
> Table 1. Comparison results between BOORL and baselines in Antmaze tasks with the normalized score metric, averaged over five random seeds with standard deviation.
>
> **W2: Deviate from the original results for PEX.**
>
> **A for W2:**
> In the offline-to-online setting, we aim to have a swift and robust improvement for the offline policy during the online phase.
> For this reason, we present the performance of the online fine-tuning policy of 200k steps, while the original PEX paper presents results of 1M steps.
> We provide the comparative results of 1M steps in Appendix G. As shown in the result, PEX's performance aligns with the results of the original paper, and BOORL significantly outperforms PEX.
>
> **W3: The legend in Figure 2 can be put to the top of two images.**
>
> **A for W3:**
> Thanks for your suggestions; we have updated the figures in the revised version.

---

> > ### Author Response · Authors · 2023-11-20
> > **Response to Reviewer JwaT (Part 2/2)**
> >
> > **Q1: Compare BOORL with additional baselines in the offline-to-online setting.**
> >
> > **A for Q1:**
> > As suggested, we compare BOORL with the SOTA baselines, Reincarnating RL and InAC.
> > The experimental results in the following table show that BOORL outperforms these two baselines. Reincarnating RL proposes a decayed behavior cloning constraint (or cross-entropy loss in discrete action space), but the choice of the scheduling of the decay process can greatly affect the performance. InAC provides strong offline performance, but there is no explicit mechanism for accelerating online fine-tuning.
> >
> >
> > | | BOORL | Reincarnating RL | InAC |
> > | :---: | :---: | :---: | :---: |
> > | hopper-random-v2 | 8.8$\rightarrow$**75.7** | 8.1$\rightarrow$8.6 | 32.0$\rightarrow$35.1 |
> > | hopper-medium-v2 | 61.9$\rightarrow$**109.8** | 60.7$\rightarrow$93.0 | 60.3$\rightarrow$80.9 |
> > | hopper-medium-replay-v2 | 75.7$\rightarrow$**111.1** | 73.1$\rightarrow$73.3 | 92.1$\rightarrow$97.8 |
> > | hopper-medium-expert-v2 | 89.0$\rightarrow$103.4 | 87.9$\rightarrow$**111.0** | 93.8$\rightarrow$93.3 |
> > | hopper-expert-v2 | 111.5$\rightarrow$**109.2** | 101.5$\rightarrow$106.1 | 103.4$\rightarrow$**108.7** |
> > | walker2d-random-v2 | 4.8$\rightarrow$**93.6** | 4.6$\rightarrow$6.9 | 2.0$\rightarrow$8.5 |
> > | walker2d-medium-v2 | 83.6$\rightarrow$**107.7** | 81.2$\rightarrow$95.0 | 82.7$\rightarrow$89.7 |
> > | walker2d-medium-replay-v2 | 69.1$\rightarrow$**114.4** | 67.8$\rightarrow$92.0 | 69.8$\rightarrow$95.5 |
> > | walker2d-medium-expert-v2 | 110.8$\rightarrow$**116.2** | 108.9$\rightarrow$111.1 | 109.0$\rightarrow$112.2 |
> > | walker2d-expert-v2 | 110.0$\rightarrow$**118.6** | 109.7$\rightarrow$110.8 | 110.6$\rightarrow$110.9 |
> > | halfcheetah-random-v2 | 10.7$\rightarrow$**97.7** | 8.9$\rightarrow$43.6 | 9.5$\rightarrow$42.1 |
> > | halfcheetah-medium-v2 | 47.9$\rightarrow$**98.7** | 46.0$\rightarrow$65.5 | 48.3$\rightarrow$63.8 |
> > | halfcheetah-medium-replay-v2 | 44.5$\rightarrow$**91.5** | 45.2$\rightarrow$60.2 | 44.3$\rightarrow$55.6 |
> > | halfcheetah-medium-expert-v2 | 77.7$\rightarrow$**97.9** | 78.4$\rightarrow$92.6 | 83.5$\rightarrow$92.9 |
> > | halfcheetah-expert-v2 | 97.5$\rightarrow$**98.4** | 96.7$\rightarrow$**97.8** | 93.6$\rightarrow$92.4 |
> > | $\delta_{\rm sum}(0.2\rm M)$ | **540.4** | 189.7 | 144.5 |
> >
> > Table 1. Comparison results between BOORL and additional baselines with the normalized score metric, averaged over five random seeds with standard deviation.
> >
> > **Q2: Why does BOORL have a different value at step 0 in some tasks?**
> >
> > **A for Q2:**
> > The main reason is that we adopt an ensemble method.
> > During the offline phase, we train $N$ policies and Q-value networks, and at each step, we select a policy randomly to generate an action. Each policy and Q-value network is trained on different samples due to the use of masks (i.e., bootstrapping). Therefore, our method has a different starting point for some tasks compared to standard offline methods. The performance of our method reduces to the same performance as TD3+BC at step 0 if we set $N= 1$.
> >
> >
> > Thanks again for your supportive comments.
> > We sincerely hope our response has cleared your concerns, and we look forward to more discussions.

---

> > > ### Comment · Reviewer_pmMK · 2023-11-22
> > >
> > > Thank you for the detailed response and the new results. I just have one more minor comment: as the other review points out, I believe a more proper citation for the 50/50 sampling citation might be [Ross et al., 2012] other than [Ball et al., 2023], if we just look at the model-free setting, [Song et al., 2022] proposed the exactly same scheme before [Ball et al., 2023].

---

> ### Author Response · Authors · 2023-11-22
> **Response to Reviewer pmMK**
>
> Thanks for your valuable comment. We have corrected it in the revised version.

---

> ### Author Response · Authors · 2023-11-22
> **Looking forward to further comments!**
>
> Dear Reviewer,
>
> We have updated our supplementary experimental results with new baselines.
> We also add the additional explanation of our algorithm and experiments.
> We are wondering if our response and revision have cleared your concerns.
> We would appreciate it if you could kindly let us know whether you have any other questions.
> We are looking forward to comments that can further improve our current manuscript.
> Thanks!
>
> Best regards,
>
> The Authors

---

### Official Review · Reviewer_pmMK · 2023-11-01

**Soundness:** 3 good
**Presentation:** 3 good
**Contribution:** 3 good
**Rating:** 6
**Confidence:** 3

**Summary:**

The paper prove a new algorithm for the offline to online RL problem by just running Thompson Sampling on both the offline and online stages. The theoretical algorithm shows that the algorithm has low Bayesian regret during the online stage, regardless of the coverage of the offline dataset. Based on the theory results, the paper proposes a practical version of the Thompson Sampling algorithm by bootstrapping from a distribution of the neural networks, and the experiments on the standard benchmarks indeed improves upon the previous algorithms in the same setting.

**Strengths:**

1. The paper performs an extensive comparison with relevant baselines, and the empirical results indeed demonstrate the proposed algorithm outperforms the other baselines.

2. The paper also includes a thorough ablation study.

3. The paper also includes a proof-of-concept experiment for the theory part, which better improve the credibility of the theory.

**Weaknesses:**

(Some of the points are questions rather than weaknesses).

1. The presentation of the theory results could be improved. Some details are left out and some indication of the theory results could be better explained. For example, the theory algorithm that is used to give the results in Thm 3.2 (and generate the results of Fig. 2) is not given so it is a little bit hard to contextualize the results. Is the algorithm running TS-LCB in the offline stage and switch to TS in the online stage?

2. From my understanding, the proposition 3.3 is trying to argue that using UCB in online stage will cause the performance drop? I am not sure that one-step suboptimality corresponds exactly to the "performance drop".

3. It would be helpful is any explanation why TS is performing better than theory in Fig. 2.

4. At the ending remark of the theory section, the paper mentions that [Song et al., 2022] benefits only when offline data has sufficient coverage, but to my best knowledge it seems that [Song et al., 2022] indeed requires and benefit from sufficient coverage from offline data. So to my understanding the current paper is achieving a best-of-both-world (kind of, not exactly) results of [Xie et al, 2021] and [Song et al., 2022].

5. The current coverage is not the tightest in the linear case. Is the tightest coverage (as in [1]) applicable in the current analysis?

6. Some indexing on $h$ seems to be off in eq 3. Also in eq 4, are the linear features not $h$-dependent?

### References
[1] Zanette, Andrea, Martin J. Wainwright, and Emma Brunskill. "Provable benefits of actor-critic methods for offline reinforcement learning." Advances in neural information processing systems 34 (2021): 13626-13640.

**Questions:**

See above

---

> ### Author Response · Authors · 2023-11-20
> **Response to Reviewer pmMK**
>
> We thank the Reviewer for the constructive comments and provide clarification to your concerns as follows. We would appreciate it if you have any further feedback.
>
> **W1: The theory algorithm that is used to give the results in Thm 3.2 (and generate the results of Fig. 2) is not given, is the algorithm running TS-LCB in the offline stage and switch to TS in the online stage?**
>
> **A for W1:**
> We thank the Reviewer for pointing this out. TS uses the same algorithm over the two stages, using all the data available to update its posterior and samples from its posterior to generate a policy during execution. In the offline phase, TS uses all offline data to update its posterior, while in the online phase, TS uses both offline data and online data to update its posterior. Others are kept the same as the vanilla TS algorithm.
>
>
> **W2: Is the Proposition 3.3 trying to argue that using UCB in online stage will cause the performance drop? and How the one-step suboptimality corresponds exactly to the performance drop?**
>
> **A for W2:**
> Proposition 3.3 is used to show that there is no per-episode performance guarantee for UCB (but a cumulative regret guarantee) due to optimistic exploration, and the performance for one episode can be arbitrarily worse when the online phase starts. On the contrary, offline learned policy has an $O(1/\sqrt{N})$ per-episode performance guarantee. Therefore, Proposition 3.3 explains why changing the offline algorithm to an online one can lead to a performance drop at the early stage.
>
> **W3: Why TS is performing better than theory in Figure 2?**
>
> **A for W3:**
> The main reason is that the theory only provides an **upper bound** for the regret. There are other factors, like the choice of the priors, that can affect the overall performance of TS.
>
>
> **W4: The statement at the ending remark of the theory section.**
>
> **A for W4:**
> We thank the Reviewer for pointing this out. More precisely, Xie et al., 2021 analyze the benefit of online exploration when offline data only has partial coverage, while Song et al., 2022 propose Hy-Q, which has an oracle property ($O(\sqrt{T})$ regret compared with **any** policy $\pi_e$) and is computationally efficient by incorporating offline data. Unlike the previous results, our result shows that using offline data with full coverage can improve over the $O(\sqrt{T})$ online regret bound by adopting a Bayesian method. We made this point more clear in the revised version.
>
> **W5: Is the tightest coverage applicable in the current analysis?**
>
> **A for W5:**
> Yes, our analysis is based on a general information-theoretic argument, independent of the notion of coverage. Our analysis generally applies to various notions of coverage coefficient, including [1,2].
>
> **W6: In Equation 3 the index for $h$ are off. In Equation 4, are the linear features not $h$-dependent?**
>
> **A for W6:**
> We thank the Reviewer for pointing it out, and we have corrected it in the revised version. A $h$-independent feature is usually assumed as in previous linear MDP literature [3,4]. We can concatenate $h$-dependent features for all $H$ steps to get an $h$-independent one so that $h$-dependent features are a special case of our definition.
>
> Thanks again for the valuable comments.
> We hope our response has cleared your concerns.
> We are looking forward to more discussion.
>
>
> [1] Zanette, Andrea, Martin J. Wainwright, and Emma Brunskill. "Provable benefits of actor-critic methods for offline reinforcement learning." Advances in neural information processing systems 34 (2021): 13626-13640.
>
> [2] Song, Yuda, et al. "Hybrid rl: Using both offline and online data can make rl efficient." arXiv preprint arXiv:2210.06718 (2022).
>
> [3] Jin, Chi, et al. "Provably efficient reinforcement learning with linear function approximation." Conference on Learning Theory. PMLR, 2020.
>
> [4] Cai, Qi, et al. "Provably efficient exploration in policy optimization." International Conference on Machine Learning. PMLR, 2020.

---

> > ### Author Response · Authors · 2023-11-22
> > **Looking forward to further comments!**
> >
> > Dear Reviewer,
> >
> > We have added the additional explanation of our algorithm, theory and experiments .
> > We are wondering if our response and revision have cleared your concerns.
> > We would appreciate it if you could kindly let us know whether you have any other questions.
> > We are looking forward to comments that can further improve our current manuscript.
> > Thanks!
> >
> > Best regards,
> >
> > The Authors

---

### Official Review · Reviewer_WzDy · 2023-11-02

**Soundness:** 3 good
**Presentation:** 3 good
**Contribution:** 3 good
**Rating:** 6
**Confidence:** 4

**Summary:**

This paper introduces a streamlined approach to transitioning from offline to online reinforcement learning (RL) through posterior sampling, eliminating the need for explicit pessimism or optimism. An information-theoretic guarantee for regret is presented. For practical algorithms, in the offline phase, an ensemble of policies is trained with bootstrap mask; in the online phase, at each step a policy is sampled from the ensemble to perform actions, and the collected data is used to update each policy network in the ensemble. The proposed method demonstrates competitive performance when benchmarked against existing algorithms.

**Strengths:**

- The explicit identification of the finetuning dilemma in offline-to-online setting is commendable.
- The streamlined Bayesian formulation is novel.
- The information-theoretic analysis on the regret bound is mostly clear and easy-to-understand.

**Weaknesses:**

- In Section 4's discussion of the replay buffer, the authors employ a symmetric sampling design, a technique previously proposed and validated in multiple prior works, e.g., [BSKL23] and [Ross et al., 2012]. It is essential to ensure that these references are comprehensively cited to acknowledge the contributions they provide and give readers a better context.
- It looks like the information ratio $\Gamma_t$ lacks of a formal definition in the paper. If so, please include this in the revision.
- Figure 2 / Appendix E: the experimental setting is slightly unclear. For UCB and LCB, what is the algorithm applied in the offline phase? Wouldn't a fair comparison for TS be using LCB at the offline phase and UCB at the online phase? The authors should also explicitly state the bandit setting (e.g., the distribution arm probabilities) for people to replicate the experiments.
- Line 4 of section 3.1: information pain --> information gain.
- Please include additional implementation specs for each experiment in the revision, e.g., the algorithm/approximation used for posterior updates, practical methods used for mutual information computation, etc.
- Why do ODT and Off2On have zero score on Antmaze tasks?
- The regret is for linear MDP. Could the author provide some discussion or proof sketch for the nonlinear case?
- A naive extension would be to use pessimistic TS (cf., [A23]) in the offline phase + optimistic TS (cf., [HZHS23]) in the online phase. Would the same analysis framework apply?

[Ross et al., 2012] Ross, Stephane, and J. Andrew Bagnell. "Agnostic system identification for model-based reinforcement learning." arXiv preprint arXiv:1203.1007 (2012).
[BSKL23] Ball, Philip J., Laura Smith, Ilya Kostrikov, and Sergey Levine. "Efficient online reinforcement learning with offline data." ICLR 2023.
[HZHS23] Hu, Bingshan, Tianyue H. Zhang, Nidhi Hegde, and Mark Schmidt. "Optimistic Thompson Sampling-based algorithms for episodic reinforcement learning." In Uncertainty in Artificial Intelligence, pp. 890-899. PMLR, 2023.
[A23] Anonymous. Posterior Sampling via Langevin Monte Carlo for Offline Reinforcement Learning. https://openreview.net/forum?id=WwCirclMvl.

**Questions:**

Please address the concerns in the weakness part.
I am happy to raise my score if the authors provide further feedback.

---

> ### Author Response · Authors · 2023-11-20
> **Response to Reviewer WzDy**
>
> Dear Reviewer,
>
> Thanks for your positive and insightful comments.
> We provide clarification to your concerns below. We would appreciate it if you have any further feedback.
>
> **W1: Explicitly citing the literature.**
>
> **A for W1:**
> Thanks for your comment. We have corrected it in the modified version.
>
> **W2:  The formal definition of the information ratio.**
>
> **A for W2:**
> We thank the Reviewer for pointing this out. The information ratio $\Gamma_t$ is the ratio between the suboptimality and the information gain [6]. We have added a concrete definition for $\Gamma_t$ in the Definition 2.2 in the revised version.
>
> **W3.1: The experimental setting in Figure 2 / Appendix E. What is the algorithm applied in the offline phase for UCB and LCB? Wouldn't a fair comparison for TS be using LCB in the offline phase and UCB in the online phase?**
>
> **A for W3.1:**
>
> We would like to clarify that there is no explicit "offline" stage for offline-to-online RL, theoretically. The choice of algorithms in the offline phase does not change the overall regret curve since it only affects the performance at step 0. For UCB/LCB algorithms, we treat all the offline data as past online experiences to calculate the statistics. For example, when we have $M$ offline data points and $N$ online data points for some arm $a$, then we use these $N+M$ data points to estimate $\mu(a)$ and $\sigma(a)$ and choose the best arm to act according to the UCB/LCB principle. We let all algorithms use both offline and online data to ensure a fair comparison.
>
>
> **W3.2: Explicitly state the bandit setting.**
>
> **A for W3.2:**
> In the bandit setting, we let each arm $a_n$ have a probability $\theta_n$ to yield a reward of 1 and a probability $1-\theta_n$ to yield a reward of 0. The parameters $\theta_n$ are i.i.d. drawn from a Beta distribution for all arms $a_n$. We use 10 arms and 1000 offline data points, and the online phase lasts for 100000 steps.
> We have included the implementation for the bandit settings in the supplementary material to ensure better replication.
>
> **W4: Typo issue.**
>
> **A for W4:**
> Thanks for your comment. We have corrected it in the revised version.
>
> **W5: Include additional implementation specs for each experiment.**
>
> **A for W5:**
> As suggested, we have added additional implementation specs in the revised version of Appendix I. Specifically, we use the proposed bootstrapping mechanism in [1] for approximate posterior updates. Mutual information is needed for the theoretical analysis rather than for the empirical implementation. There are various methods like [2] for computing approximate mutual information for theoretical verification.
>
> **W6: Why do ODT and Off2On have zero score on Antmaze tasks?**
>
> **A for W6:**
> Antmaze tasks are challenging for exploration, so it is important to use offline data properly to complete the tasks. At the offline stage, ODT is based on DT, which is known to fail on Antmaze tasks [3], while Off2On is built on CQL, which has similar issues [4].
>
>
> **W7: Could the author provide some discussion or proof sketch for the nonlinear case?**
>
> **A for W7:**
> While Section 3.2 focuses on linear MDPs, Section 3.1 provides an information-theoretic analysis generally applicable to different MDP structures. To apply for the nonlinear setting, we need to analyze the information ratio $\Gamma_t$ (defined in Definition 2.2) for nonlinear settings, and others are kept the same. In linear settings, $\Gamma$ is a constant, while it can be a function of $t$ in general settings. For example, for Gaussian kernel MDPs, $\Gamma_t=O(\log t)$ [5] and the regret bound will be increased by a factor of $log T$ correspondingly.
>
> **W8: Would the same analysis framework apply to the naive extension? (e.g., pessimistic TS in the offline phase and optimistic TS in the online phase)**
>
> **A for W8:**
>
> As discussed in W3, the choice of algorithms in the offline stage theoretically does not change the overall performance. While optimistic TS provides a nice frequentist guarantee compared to vanilla TS, it suffers from the same issue as UCB, which can lead to a significant performance drop at an early stage due to optimistic exploration.
>
>
> Thanks again for the valuable comments.
> We sincerely hope our response has addressed your concerns.

---

> > ### Author Response · Authors · 2023-11-20
> > **Reference**
> >
> > [1] Osband, Ian, et al. "Deep exploration via bootstrapped DQN." Advances in neural information processing systems 29 (2016).
> >
> > [2] Belghazi, Mohamed Ishmael, et al. "Mutual information neural estimation." International conference on machine learning. PMLR, 2018.
> >
> > [3] Kostrikov, Ilya, Ashvin Nair, and Sergey Levine. "Offline reinforcement learning with implicit q-learning." arXiv preprint arXiv:2110.06169 (2021).
> >
> > [4] Wang, Jianhao, et al. "Offline reinforcement learning with reverse model-based imagination." Advances in Neural Information Processing Systems 34 (2021): 29420-29432.
> >
> > [5] Valko, Michal, et al. "Finite-time analysis of kernelised contextual bandits." arXiv preprint arXiv:1309.6869 (2013).
> >
> > [6] Russo, Daniel, and Benjamin Van Roy. "An information-theoretic analysis of thompson sampling." The Journal of Machine Learning Research 17.1 (2016): 2442-2471.

---

> ### Author Response · Authors · 2023-11-22
> **Looking forward to further comments!**
>
> Dear Reviewer,
>
> We have added the additional explanation of our algorithm, theory and experiments .
> We are wondering if our response and revision have cleared your concerns.
> We would appreciate it if you could kindly let us know whether you have any other questions.
> We are looking forward to comments that can further improve our current manuscript.
> Thanks!
>
> Best regards,
>
> The Authors

---

### Official Review · Reviewer_x3PJ · 2023-11-04

**Soundness:** 3 good
**Presentation:** 3 good
**Contribution:** 3 good
**Rating:** 5
**Confidence:** 3

**Summary:**

The paper proposes to use a Bayesian approach to balance exploration and exploitation in the offline-to-online RL domain. Theoretical analysis shows the regret of the proposed method. experimental results show good performance compared to some popular baselines.

**Strengths:**

see question part

**Weaknesses:**

see question part

**Questions:**

The paper proposes to use a Bayesian method to balance exploration and exploitation to avoid performance drop. A distributional RL method is combined with the ensemble approach to do the Bayesian exploration. The theoretical analysis seems sound and the performance of the proposed method is good. However, I still have some concerns.
1. As we all know, the ensemble trick is helpful in improving the performance of RL methods and is widely used in practice. It seems that it is unfair to use an approach with the ensemble method to compare with other methods without the ensemble. I wondering if there is an ablation study to show the performance of an ensemble version of TD3 or TD3+BC.

2. The key part of the algorithm is not clear. Specifically, the mask m samples from a distribution M. What is the format of M and how does it initialize and update? what is the difference between the usage of m compared to the original ensemble method? In the online part, it seems that the distribution M can be seen as the priori according definition of Bayesian. However, the authors choose to use uniform distribution to choose policy, which is the same as the original ensemble method. Could the authors explain it?

---

> ### Author Response · Authors · 2023-11-20
> **Response to Reviewer x3PJ**
>
> Dear Reviewer,
>
> Thanks for your valuable comments. We have provided additional experimental results and explanations to address your concerns, and we hope the following clarifications shed light on the points you raised.
>
>
> **C1: Is there an ablation study to show the performance of an ensemble version of TD3 or TD3+BC?**
>
> **A for C1:**
> As suggested, we conduct ablation studies to investigate the performance of the ensemble versions of TD3 and TD3+BC.
> Specifically, we use an ensemble version of TD3+BC during the offline phase and adopt the ensemble versions of TD3 and TD3+BC during the online fine-tuning phase.
> The experimental results of the following table show that BOORL significantly outperforms these baselines. Different from these baselines, BOORL adopts a bootstrapping mechanism where each agent uses a different dataset sampled uniformly with replacement from the original dataset. This mechanism is critical in improving offline-to-online performance, as shown in the ablation study in Appendix H.1.
>
>
> | | BOORL | TD3 (Ensemble) | TD3+BC (Ensemble) |
> | :---: | :---: | :---: | :---: |
> | hopper-random-v2 | **75.7$\pm$1.3** | 20.2$\pm$3.4 | 8.0$\pm$3.6 |
> | hopper-medium-v2 | **109.8$\pm$1.6** | 93.2$\pm$1.4 | 54.2$\pm$1.6 |
> | hopper-medium-replay-v2 | **111.1$\pm$0.3** | 98.0$\pm$0.7 | 66.5$\pm$2.6 |
> | walker2d-random-v2 | **93.6$\pm$0.4** | 2.2$\pm$0.2 | 1.7$\pm$0.1 |
> | walker2d-medium-v2 | **107.7$\pm$0.5** | 78.2$\pm$2.2 | 81.8$\pm$2.4 |
> | walker2d-medium-replay-v2 | **114.4$\pm$0.9** | 79.3$\pm$1.2 | 82.6$\pm$1.8 |
> | halfcheetah-random-v2 | **97.7$\pm$1.1** | 65.9$\pm$0.5 | 38.7$\pm$0.2 |
> | halfcheetah-medium-v2 | **98.7$\pm$0.3** | 70.5$\pm$1.0 | 51.9$\pm$0.5 |
> | halfcheetah-medium-replay-v2 | **91.5$\pm$0.9** | 71.1$\pm$1.1 | 50.8$\pm$0.8 |
>
> Table 1. Comparison results between BOORL and the ensemble version of TD3 and TD3+BC with the normalized score metric, averaged over five random seeds with standard deviation.
>
> **C2.1: What is the format of M, and how does it initialize and update? Why use the uniform distribution to choose a policy in the online part?**
>
> **A for C2.1:**
> We process the mask distribution consistent with [1] to simulate the bootstrapping process.
> Specifically, $ M$ is a Bernoulli distribution with a parameter $ p$, and we sample mask $ m_1, m_2,..., m_N$ i.i.d. from Bernoulli ($ p$). Therefore, each mask generates a bootstrapped dataset (i.e., iid sampling with replacement) from the original one. We draw new masks for each new online data point when it is added to the replay buffer, but the masks are not updated after that.
> Each policy trained with one bootstrapped dataset can be seen as a sample from the posterior [1]. Therefore
>  uniformly choosing a policy in the sampled set of $N$ policies is an approximation of sampling from the posterior from a Bayesian point of view.
>
>
>
>
> **C2.2: What is the difference between the usage of m compared to the original ensemble method?**
>
> **A for C2.2:**
>
> The original ensemble method uses the same dataset and different initialization of neural networks to train ensembled policies. While this method can reduce variance, it does not properly reflect epistemic uncertainty [3]. The usage of the mask is derived from the bootstrapping method where each policy uses its own dataset, i.i.d. sampled with replacement from the original dataset. This helps diversify the policies and can reflect the posterior properly.
>
> Thanks again for the valuable comments.
> We sincerely hope our additional explanation of the experimental setup has cleared the concern.
> More comments on further improving the presentation are welcomed.
>
> [1] Osband, Ian, et al. "Deep exploration via bootstrapped DQN." Advances in neural information processing systems 29 (2016).
>
> [2] Chen, Xinyue, et al. "Randomized Ensembled Double Q-Learning: Learning Fast Without a Model." International Conference on Learning Representations. 2020.
>
> [3] Chua, Kurtland, et al. "Deep reinforcement learning in a handful of trials using probabilistic dynamics models." Advances in neural information processing systems 31 (2018).

---

> ### Author Response · Authors · 2023-11-22
> **Looking forward to further comments!**
>
> Dear Reviewer,
>
> We have updated our supplementary experimental results in the ensemble version of TD3 and TD3+BC.
> We also add an additional explanation of our algorithm and experiments.
> We are wondering if our response and revision have cleared your concerns.
> We would appreciate it if you could kindly let us know whether you have any other questions.
> We are looking forward to comments that can further improve our current manuscript.
> Thanks!
>
> Best regards,
>
> The Authors

---

### Author Response · Authors · 2023-11-20
**Rebuttal Summary**

Dear Reviewers,

We thank all the reviewers for their constructive and valuable comments. We are encouraged to learn that many found our work "commendable" in identifying the off-to-on dilemma, "novel" in the proposed streamline TS approach, "credible" in the theory, and "clear and easy to follow" in writing. We genuinely appreciate these positive remarks.

We provide experimental results and clarification to the concerns raised about the details of the theory and bandit experiment, the comparison with other baselines, and the details of the experiments. In response:

1. **[Theoretical Explanation]** We give a formal definition for the information ratio, a detailed comparison with prior works and give a more detailed explanation for the offline-to-online transition in bandit settings.

2. **[More Baselines]** We conduct additional experiments and compare our methods to the ensemble version of TD3 and TD3+BC and two additional baselines: Reincarnating RL and InAC.

3. **[Details for the Bandit Experiment]** We add more details for the bandit experiment, including offline data usage for UCB/LCB, our algorithm, and the experiment specs. We also include the implementation for the bandit setting in the supplementary material.

We sincerely hope these updates and clarifications address the Reviewers' concerns. We welcome further discussion and suggestions for improving the paper.

---

### Meta-Review · Area_Chair_NHDn · 2023-12-09

**Metareview:**

In this paper, the authors study a form of the offline-online RL problem (in the finite-horizon setting) in which the goal is to control the Bayesian regret in the online phase. They propose an algorithm that just runs Thompson Sampling (TS) in the online phase using the offline data as a prior (online data is used to update the posterior). They derive a (Bayesian) regret bound for this algorithm in linear MDPs and provide theoretical evidence that it is superior to simply acting pessimistic (LCB) or optimistic (UCB) in the online phase. They propose a practical version of their TS algorithm that is based on bootstrapping from a distribution of the neural networks. Finally, they empirically evaluate their algorithm and compare it with existing methods using standard benchmarks.

The problem of offline-online RL studied in the paper is important. Of course, this is only a specific formulation of this problem suitable for certain settings. The paper is a good balance of theoretical analysis, algorithmic implementation, and empirical evaluation (good amount of  comparison with relevant baselines and ablation study).

However, the novelty of the paper is quite limited. The theoretical findings are not surprising. It is well-understood that neither pessimism nor optimism is the solution to the offline-online RL problem. Moreover, the theoretical derivations are quite standard, the proofs can trivially be derived from the existing results. The derivation of the practical algorithm also follows standard steps.

Minor Comments:
- The presentation of the theoretical results and discussion about them (specifically the comparison with UCB and LCB, mainly UCB) could be improved. Moreover, the notation (t,k,h) could be used better, more consistently and in a less confusing way.
- It is misleading to call the problem studied in the paper simply as offline-online RL. As I mentioned above, this is a specific form of the offline-online RL problem in which the goal is to control the Bayesian regret in the online phase. Obviously, this setting is suitable for some problems and not sufficient for other.

**Justification For Why Not Higher Score:**

However, the novelty of the paper is quite limited. The theoretical findings are not surprising. It is well-understood that neither pessimism nor optimism is the solution to the offline-online RL problem. Moreover, the theoretical derivations are quite standard, the proofs can trivially be derived from the existing results. The derivation of the practical algorithm also follows standard steps.

**Justification For Why Not Lower Score:**

N/A

---

### Decision · Program_Chairs · 2024-01-16

Reject